# Oxophilic Ce single atoms-triggered active sites reverse for superior alkaline hydrogen evolution

Fengyi Shen [1,3], Zhihao Zhang[2,3], Zhe Wang[1,3], Hao Ren[1], Xinhu Liang[1], Zengjian Cai[1], Shitu Yang[1], Guodong Sun[1], Yanan Cao[1], Xiaoxin Yang[1], Mingzhen Hu [1,2] ✉, Zhengping Hao [2] ✉ & Kebin Zhou [1,2] ✉

The state-of-the-art alkaline hydrogen evolution catalyst of united ruthenium single atoms and small ruthenium nanoparticles has sparked considerable research interest. However, it remains a serious problem that hydrogen evolution primarily proceeds on the less active ruthenium single atoms instead of the more efficient small ruthenium nanoparticles in the catalyst, hence largely falling short of its full activity potential. Here, we report that by combining highly oxophilic cerium single atoms and fully-exposed ruthenium nanoclusters on a nitrogen functionalized carbon support, the alkaline hydrogen evolution centers are facilely reversed to the more active ruthenium nanoclusters driven by the strong oxophilicity of cerium, which significantly improves the hydrogen evolution activity of the catalyst with its mass activity up to $-10.1\,A\,mg^{-1}$ at $-0.05\,V$. This finding is expected to shed new light on developing more efficient alkaline hydrogen evolution catalyst by rational regulation of the active centers for hydrogen evolution.

Producing hydrogen via water electrolysis has been regarded as one of the most promising approaches to mediate the climate change and the global energy crisis[1,2]. In practical applications, alkaline water electrolysis and proton exchange membrane (PEM)-based water electrolysis in acid electrolyte both have their advantages for hydrogen production[3]. In a typical PEM system, a PEM was used as solid electrolyte by means of which proton could be facilely transferred to the cathode[4], enabling fast hydrogen evolution kinetics. However, unlike direct proton supply pattern in acid hydrogen evolution reaction (HER), the proton was provided by water dissociation during the alkaline HER (Volmer step, Eq. 1)[5–7]. The substantial energy barrier for scissoring the OH-H bond and sluggish supply of proton inevitably impeded the reaction rate of alkaline HER[8–10]. As a result, even for the benchmark commercial Pt/C electrocatalyst, about two orders of magnitude reduction in hydrogen evolution activity was commonly identified when used for alkaline HER[11]. Therefore, promoting the

water dissociation capability of electrocatalyst is of paramount importance to boost the alkaline hydrogen evolution kinetics.

$$H_2O + e^- \rightarrow H_{ad} + OH^- (\text{Volmer step}) \qquad (1)$$

$$H_2O + H_{ad} + e^- \rightarrow H_2 + OH^- (\text{Heyrovsky step}) \qquad (2)$$

$$H_{ad} + H_{ad} \rightarrow H_2 (\text{Tafel step}) \qquad (3)$$

In the past decades, ruthenium (Ru) has shown great potential to substitute the platinum (Pt) for alkaline HER by virtue of two-fold: (i) the metal price of Ru (ca. 15 $ g$^{-1}$, Sept 2023) was less than half that of the Pt (ca. 34 $ g$^{-1}$, Sept 2023); and (ii) the energy barrier for water dissociation over Ru was much lower than that over Pt[12–14]. To maximize Ru metal efficiency, Ru single atom catalysts were firstly

[1]School of Chemical Sciences, University of Chinese Academy of Sciences, Beijing 100049, PR China. [2]National Engineering Laboratory for VOCs Pollution Control Material & Technology, Research Center for Environmental Material and Pollution Control Technology, University of Chinese Academy of Sciences, Beijing 100049, PR China. [3]These authors contributed equally: Fengyi Shen, Zhihao Zhang, Zhe Wang. ✉e-mail: humingzhen12@ucas.ac.cn; zphao@ucas.ac.cn; kbzhou@ucas.ac.cn

employed for the alkaline HER[15-17]. Nevertheless, the Ru single atoms exhibited very low reactivity for water dissociation, which significantly impeded their alkaline HER activities.

Most recently, the united catalyst of Ru single atoms and small Ru nanoparticles ($Ru_1$-$Ru_n$) has garnered tremendous research interest because of its attractive alkaline HER activity[18-24]. Theoretical calculations in Fig. S1 suggested that water dissociation over the $Ru_1$-$Ru_n$ catalyst was both thermodynamically and kinetically favorable to proceed via a "$Ru_1H$-$Ru_nOH$ route" with the OH of water adsorbed on the $Ru_n$ side and the H of water adsorbed on the $Ru_1$ side of the catalyst. Because of this, the ultimate hydrogen evolution primarily occurred on the $Ru_1$ side of the $Ru_1$-$Ru_n$ catalyst. However, this kind of hydrogen evolution mode had a major disadvantage that the $Ru_1$ sites were less active for hydrogen evolution than the $Ru_n$ sites as suggested by the theoretical calculations in Fig. S2, which meant that the alkaline HER activity of the $Ru_1$-$Ru_n$ catalyst largely fell short of its full activity potential. As such, it is highly desired to reverse the hydrogen evolution centers from the less reactive Ru single atom side to the more reactive $Ru_n$ nanocluster side in the $Ru_1$-$Ru_n$ catalyst to expedite the alkaline HER efficiency, which is urgently awaited to be explored.

Herein, we report that by uniting oxophilic cerium (Ce) single atoms and fully-exposed $Ru_n$ nanoclusters on a N functionalized carbon support, the alkaline hydrogen evolution centers were facilely reversed to the more efficient $Ru_n$ nanoclusters. The driving force for the active site reverse was the strong oxophilicity of Ce by means of which the OH produced by water dissociation was selectively bonded with the Ce single atoms while the H was moderately adsorbed on the fully-exposed $Ru_n$ nanoclusters (Fig. 1), thereby favoring superior hydrogen evolution performance. Furthermore, water dissociation was also significantly promoted over the $Ce_1$-$Ru_n$/NC catalyst, benefiting from the strong synergies between the highly oxophilic Ce single atoms and the fully-exposed $Ru_n$ nanoclusters. Consequently, the alkaline HER activity of the $Ce_1$-$Ru_n$/NC catalyst was largely improved relative to that of the $Ru_1$-$Ru_n$/NC catalyst.

## Results and discussion

### Synthesis and characterization of the $Ce_1$-$Ru_n$/NC catalyst

In preparation, a N modification procedure was firstly conducted to functionalize the XC-72 carbon (denoted as NC support, please see methods part for details). Ce precursors were then introduced to the NC support by impregnation and allowed for further hydrogen reduction at 700 °C. After the treatment of acid etching, the NC supported Ce single atoms were acquired ($Ce_1$/NC). To synthesize the united catalyst of Ce single atoms and small $Ru_n$ nanoclusters on the NC support ($Ce_1$-$Ru_n$/NC), Ru precursors were then impregnated onto the $Ce_1$/NC and reduced by hydrogen at 250 °C. As a control, the $Ru_n$ nanoclusters were prepared on the NC support (denoted as $Ru_n$/NC) using a similar method. The detailed synthetic process was also

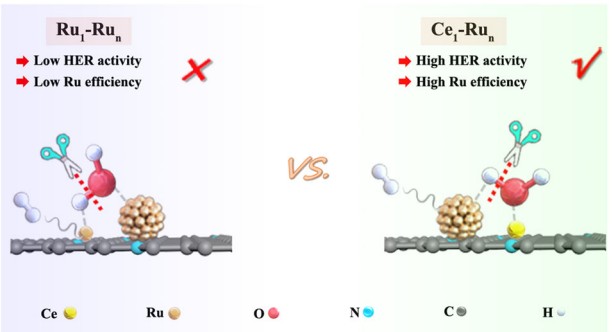

**Fig. 1 | Reaction schemes for alkaline HER over the $Ru_1$-$Ru_n$ and $Ce_1$-$Ru_n$.** Schematic illustration of the distinct alkaline hydrogen evolution modes over the $Ru_1$-$Ru_n$ catalyst and over the $Ce_1$-$Ru_n$ catalyst, respectively.

displayed in Fig. S3 and in the methods part, respectively. In addition, we have further provided the schematic models of the $Ce_1$/NC, $Ru_n$/NC, and $Ce_1$-$Ru_n$/NC catalysts in Fig. 2a, Fig. 2h, and Fig. 2o, respectively.

The loading amount of Ce in the $Ce_1$/NC catalyst was measured to be 0.03 wt% by an inductively coupled plasma optical emission spectrometer (ICP-OES) as displayed in Table S1. Due to the low Ce loadings, only the broad X-ray diffraction (XRD) peaks of carbon support[25] were observed for the $Ce_1$/NC catalyst as displayed in Fig. S4. Microscopic measurement via the high angle annular dark field scanning transmission electron microscopy (HAADF-STEM) technique of the $Ce_1$/NC catalyst in Fig. S5–7 ruled out the nanoparticles in it. As was further revealed by the aberration-corrected HAADF-STEM (AC HAADF-STEM) measurement, Ce species were atomically dispersed in the $Ce_1$/NC catalyst (Fig. 2b–d). Corresponding energy-dispersive X-ray spectrometry (EDS) elementary mapping images of the $Ce_1$/NC catalyst in Fig. 2e–g also confirmed the uniform dispersion of Ce species in it.

Likewise, we have characterized the $Ru_n$/NC and the $Ce_1$-$Ru_n$/NC catalysts via the microscopic techniques. As presented by the HAADF-STEM images of the $Ru_n$/NC catalyst in Fig. 2i–k, fully-exposed $Ru_n$ nanoclusters with an average particle size of $1.1 \pm 0.3$ nm were evenly dispersed on its NC support. The EDS elementary mapping images of a small nanocluster in the $Ru_n$/NC catalyst (Fig. 2l–n) revealed its Ru element nature. The AC HAADF-STEM images of the $Ce_1$-$Ru_n$/NC catalyst in Fig. 2p–r showed that the mean particle size of the fully-exposed Ru nanoclusters in it was $1.0 \pm 0.2$ nm. Corresponding EDS elementary mapping images of the $Ce_1$-$Ru_n$/NC catalyst in Fig. 2s–u further indicated that the Ru element signal was concentrated on the small nanoclusters while the Ce element signal was highly overlapped with the distribution of single atoms, which suggestd the copresence of Ru nanoclusters and Ce single atoms in it. No diffraction peaks of Ru were identified for both of the $Ru_n$/NC catalyst and the $Ce_1$-$Ru_n$/NC catalyst as shown in Fig. S8 because of the ultrasmall Ru nanoclusters in them. The Ru loading amounts of the $Ce_1$-$Ru_n$/NC catalyst and the $Ru_n$/NC catalyst were both about 1 wt% (Table S1) while the Ce weight loading in the $Ce_1$-$Ru_n$/NC was 0.03 wt% as determined by the ICP-OES measurements.

The coordination environment of Ru and Ce in the $Ce_1$/NC, $Ru_n$/NC, and $Ce_1$-$Ru_n$/NC catalysts were further examined by the X-ray absorption fine structure (XAFS) measurements. Fig. 3a shows the near-edge XAFS spectra (XANES) at Ru K-edge of the $Ce_1$-$Ru_n$/NC catalyst, $Ru_n$/NC catalyst and reference Ru foil and $RuO_2$. It was displayed by the enlarged Ru K-edge XANES spectra in the inset of Fig. 3a that the edge absorption energies of the $Ce_1$-$Ru_n$/NC catalyst and $Ru_n$/NC catalyst were between Ru foil and $RuO_2$, which demonstrated their Ru oxidation states were between 0 and +4.

On the other hand, as suggested by the best fitted Fourier transforms of the Ru K-edge extended XAFS (EXAFS) spectrum in Table S2 (corresponding fitting curves were shown in Fig. S9, 10), the mean Ru-Ru coordination number of the $Ru_n$/NC catalyst was about 2, which was markedly lower than that of a pure 1 nm $Ru_n$ nanocluster (6.5, $n = 13$) as displayed in Fig. S11. This result implied the copresence of Ru single atoms and $Ru_n$ nanoclusters in the $Ru_n$/NC catalyst because the Ru single atoms were solely coordinated with N and thus decreased the mean Ru-Ru coordination numbers of the $Ru_n$/NC catalyst. Careful aberration-corrected HAADF-STEM observation of the $Ru_n$/NC catalyst in Fig. S12 also revealed the presence of Ru single atoms in the $Ru_n$/NC catalyst. The partially positively charged Ru single atoms in the $Ru_n$/NC catalyst inevitably increased the average oxidation state of Ru, because of which the edge absorption energy of the $Ru_n$/NC catalyst showed a marked increase relative to that of the metallic Ru foil as mentioned above. It was further found in Fig. 3a that the edge energy of the $Ce_1$-$Ru_n$/NC catalyst showed a negative shift relative to the $Ru_n$/NC catalyst and thus an enhancement of the mean Ru electron density of the $Ce_1$-$Ru_n$/NC catalyst[26,27], which was possibly derived from the Ce electron donation as the only difference of Ru between the $Ce_1$-$Ru_n$/

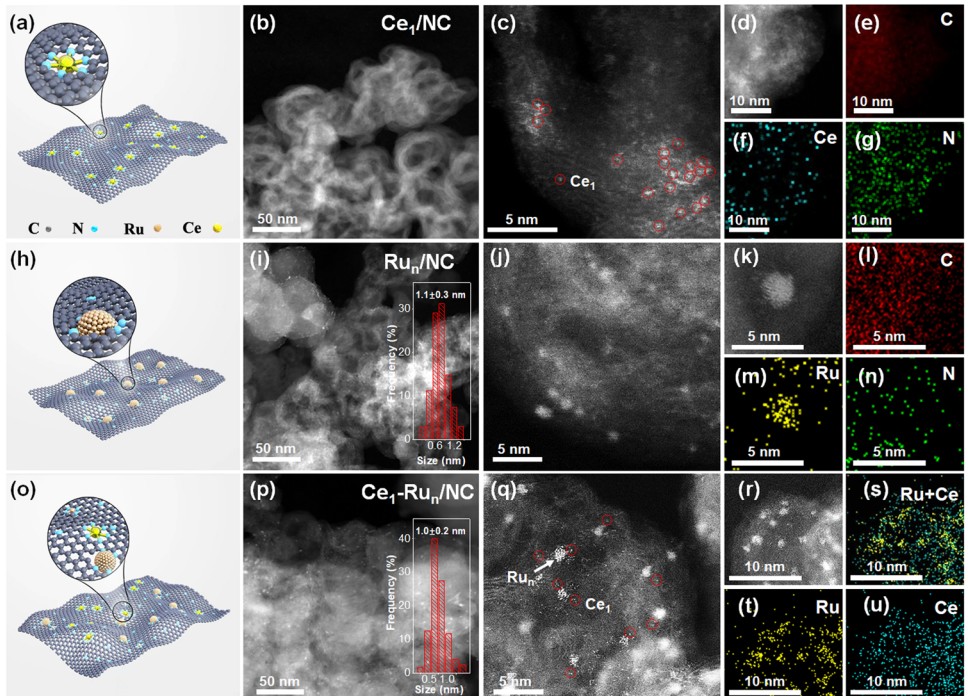

**Fig. 2 | Microscopic measurements of the Ce₁/NC, Ruₙ/NC, and Ce₁-Ruₙ/NC, respectively.** Structural models of the Ce₁/NC, Ruₙ/NC, and Ce₁-Ruₙ/NC were shown in (**a**), (**h**), and (**o**), respectively. **b**–**d** AC HAADF-STEM images, (**e**–**g**) EDS elementary mapping images of the Ce₁/NC. **i**–**k** HAADF-STEM images, (**l**–**n**) EDS elementary mapping images of the Ruₙ/NC. (**p**–**r**) AC HAADF-STEM images, (**s**–**u**) EDS elementary mapping images of the Ce₁-Ruₙ/NC. The insets of (**i**) and (**p**) are the histograms of particle size distribution for Ru nanocluster in the Ruₙ/NC and Ce₁-Ruₙ/NC, respectively.

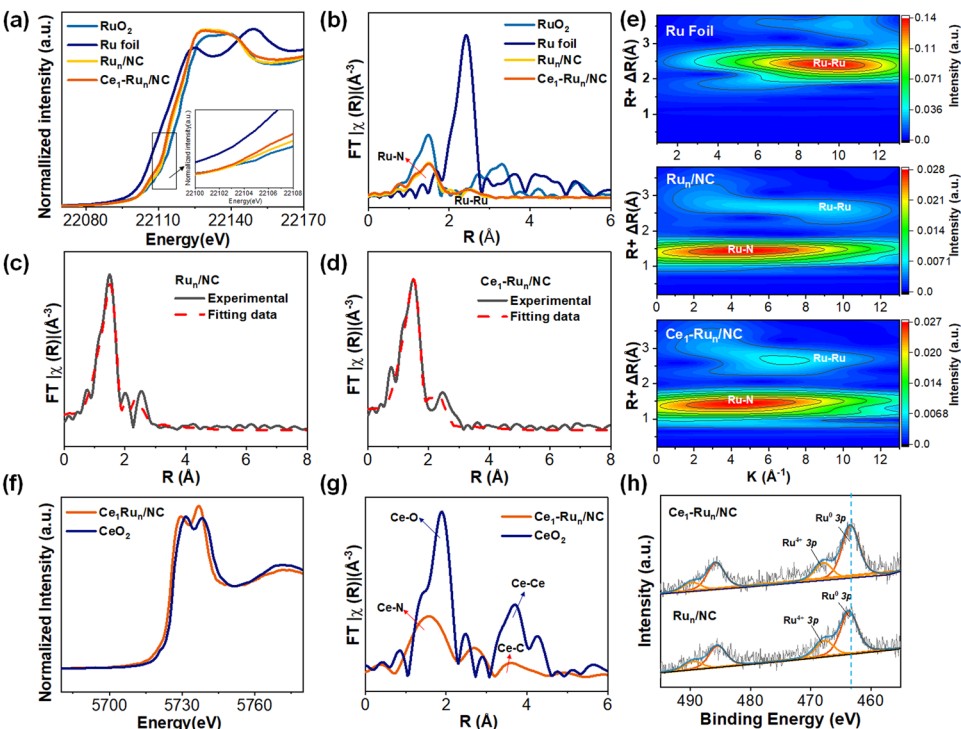

**Fig. 3 | Characterizations of the coordination and oxidation states of the Ce₁/NC, Ruₙ/NC, and Ce₁-Ruₙ/NC, respectively. a** Ru K-edge XANES spectra, (**b**) Fourier transforms of the Ru K-edge EXAFS oscillations in the R space of the Ce₁-Ruₙ/NC, Ruₙ/NC, and reference Ru foil and RuO₂. **c** and (**d**) EXAFS fitting curves of the Ce₁-Ruₙ/NC and the Ruₙ/NC. **e** Wavelet Transformation for the EXAFS signals of the Ce₁-Ruₙ/NC, Ruₙ/NC and Ru foil. **f** Ce L(III)-edge XANES spectra, (**g**) Fourier transforms of the Ce L(III)-edge EXAFS oscillations in the R space of the Ce₁-Ruₙ/NC and the reference CeO₂. **h** Ru 3p XPS spectra of the Ce₁-Ruₙ/NC and the Ruₙ/NC.

NC catalyst and the Ru$_n$/NC catalyst was the presence of Ce single atoms in the Ce$_1$-Ru$_n$/NC catalyst. To check this assumption, the Bader charge analysis was further conducted. As suggested in Fig. S13, the Ce electrons could be facilely transferred to Ru with a net electron transfer number of 0.05 and thus enhanced the mean electron density of Ru in the Ce$_1$-Ru$_n$/NC catalyst relative to that of the Ru$_n$/NC catalyst.

The Fourier transforms of the Ru K-edge EXAFS oscillations in the R space of the Ce$_1$-Ru$_n$/NC catalyst and the Ru$_n$/NC catalyst in Fig. 3b presented a main coordination peak below 2 Å, matching to the Ru-N coordination[28]. In the meanwhile, the Ru-Ru coordination peak at about 2.7 Å was also identified both for the Ce$_1$-Ru$_n$/NC catalyst and the Ru$_n$/NC catalyst. However, the peak intensity of Ru-Ru coordination was largely reduced relative to Ru-N coordination in both of the two catalysts, revealing the marked difference between Ru-Ru and Ru-N coordination numbers in them[29]. The EXAFS data fitting curves of the two catalysts in Fig. 3c, d and corresponding fitting data in Table S2 further unveiled that the Ru-N coordination ratio was largely increased compared with the Ru-Ru coordination in both of the two catalysts. In addition, the enhanced Ru-N coordination ratios in the Ce$_1$-Ru$_n$/NC and Ru$_n$/NC catalysts were also supported by the EXAFS wavelet transform results as displayed in Fig. 3e, wherein the Ru-N coordination intensity was obviously stronger than the Ru-Ru coordination. This result was understandable because the particle downsizing of Ru in the two catalysts would significantly decouple the Ru-Ru bond and strengthen the Ru-N coordination in them as was also reported previously[27].

In another respect, the Ce L$_{III}$-edge XANES and fitted EXAFS spectra of the Ce$_1$-Ru$_n$/NC catalyst and the reference CeO$_2$ in Fig. 3f, g and Fig. S14, 15 again indicated the single atom nature of Ce species in the Ce$_1$-Ru$_n$/NC catalyst, which was consistent with the above AC HAADF-STEM observation. Other than that, we have also examined the Ce-Ce coordination in the Ce$_1$-Ru$_n$/NC catalyst and the reference CeO$_2$ to further examine the existence form of Ce species in the Ce$_1$-Ru$_n$/NC catalyst. As indicated by the Fourier transforms of the Ce L$_{III}$-edge EXAFS spectra in Fig. 3g, a coordination peak at 3.8 Å was identified for the reference CeO$_2$, which was assigned to the Ce-Ce coordination by data fitting. At the same time, the Ce$_1$-Ru$_n$/NC catalyst presented a coordination peak at 3.2 Å that was markedly shorter than that of the Ce-Ce coordination. Careful data fitting suggested that this coordination peak of the Ce$_1$-Ru$_n$/NC catalyst was derived from the Ce-C contribution in the second coordination shell. To further confirm this, we have conducted the high-resolution EXAFS wavelet transform characterizations. As demonstrated in Fig. S16, the reference CeO$_2$ displayed an intensity maximum at 7.7 Å$^{-1}$ (derived from the Ce-Ce contribution) while the Ce$_1$-Ru$_n$/NC catalyst showed an intensity maximum at 2.3 Å$^{-1}$ (originated from the Ce-C contribution). This result again uncovered the absence of Ce-Ce bond in the Ce$_1$-Ru$_n$/NC catalyst.

Furthermore, the electron transfer between Ce single atoms and Ru nanoclusters in the Ce$_1$-Ru$_n$/NC catalyst was further evidenced by the Ru 3$p$ XPS spectra in Fig. 3h, wherein the Ru$^0$ binding energy peak of the Ce$_1$-Ru$_n$/NC catalyst was slightly negatively shifted compared with the Ru$_n$/NC catalyst because of the electron back-donation from Ce to Ru. The full XPS spectrum of the Ce$_1$-Ru$_n$/NC catalyst was also provided in Fig. S17 and the N 1$s$ and C 1$s$ XPS peaks of the catalyst were evidently identified. However, due to the low mass loadings of Ce in the Ce$_1$-Ru$_n$/NC catalyst, no Ce XPS peak was found in the survey XPS spectrum. Thanks to the low detection limit of the aforementioned XAFS measurement, the presence of Ce in the Ce$_1$-Ru$_n$/NC catalyst was well verified as demonstrated in Fig. 2f.

## Electrocatalytic evaluations for alkaline HER

According to reported findings[30], water dissociation on catalyst was largely dependent on corresponding OH adsorption strength. Stronger OH binding was beneficial to water molecules' polarization and elongation and commonly promoted water dissociation. Therefore, we have firstly conducted the cyclic voltammetry (CV) measurements in

1.0 M KOH solution to explore the OH adsorption strength on the Ru$_n$/NC catalyst and Ce$_1$-Ru$_n$/NC catalyst, respectively. Fig. 4a displayed that the OH-desorption peak intensity around 0.60–1.00 V[31] of the Ce$_1$-Ru$_n$/NC catalyst was much stronger than that of the Ru$_n$/NC catalyst, indicating intensified OH adsorption on the Ce$_1$-Ru$_n$/NC catalyst over the Ru$_n$/NC catalyst. In addition, the OH desorption peak of the Ce$_1$-Ru$_n$/NC catalyst also displayed a negative shift relative to that of the Ru$_n$/NC catalyst, which further verified the stronger OH adsorption on the Ce$_1$-Ru$_n$/NC catalyst than that on the Ru$_n$/NC catalyst. As a control, we have also prepared a CeO$_2$ nanoparticle integrated Ru$_n$ catalyst on the NC support (Ru$_n$-CeO$_2$/NC, detailed characterizations for the catalyst were provided in Fig. S18–22) with similar-sized Ru nanoclusters to those of the Ru$_n$/NC and Ce$_1$-Ru$_n$/NC catalysts while the Ce sites in the Ru$_n$-CeO$_2$/NC catalyst were bonded with O. It was found that the OH-desorption peak intensity of the Ru$_n$-CeO$_2$/NC catalyst was significantly decreased relative to that of the Ce$_1$-Ru$_n$/NC catalyst due to the O blocking of Ce sites. These results indicated that OH was more prone to bond with Ce single atoms of the Ce$_1$-Ru$_n$/NC catalyst as vividly depicted in Fig. 4b.

To further confirm this, we have performed the in situ Raman measurements in 1.0 M KOH electrolyte under real alkaline HER conditions (the instrument model was schematically illustrated in Fig. S23). As presented in Fig. 4c, the Raman vibration peaks at 1530 cm$^{-1}$ and 1390 cm$^{-1}$ of the Ce$_1$-Ru$_n$/NC catalyst were gradually intensified with the decrease of reaction potentials during the alkaline HER process. By contrast, the Ru$_n$/NC catalyst merely showed the vibration peaks for the D band and G band of carbon support (Fig. 4d) under the same testing conditions[32]. Since there was no record of the two Raman vibration peaks of the Ce$_1$-Ru$_n$/NC catalyst at 1530 cm$^{-1}$ and 1390 cm$^{-1}$, we have further conducted the theoretical simulations for assigning the two peaks. Through theoretical fitting of the Raman spectrum of the Ce$_1$-Ru$_n$/NC catalyst before alkaline HER test (Fig. 4e), we found that the Raman vibration peaks at 1530 cm$^{-1}$ and 1390 cm$^{-1}$ were assigned to the Ce-N stretching vibration in it but the two peaks were very weak before the alkaline HER test. Thus, it was reasonable to assume that the strengthening of the Raman vibration peaks at 1530 cm$^{-1}$ and 1390 cm$^{-1}$ was related to OH or H produced during the alkaline HER process. To this end, we have examined both OH and H effects on the Ce-N stretching vibration. Strikingly, upon introducing OH to the Ce-N sites of the Ce$_1$-Ru$_n$/NC catalyst, corresponding Raman peak intensities at 1530 cm$^{-1}$ and 1390 cm$^{-1}$ were significantly strengthened as shown in Fig. 4f, agreeing well with the in situ Raman spectra in Fig. 4c. By comparison, it was manifested in Fig. S24 that H had a negligible impact on intensifying the Ce-N stretching vibration at 1530 cm$^{-1}$ and 1390 cm$^{-1}$. Evidently, OH was more prone to bond with the Ce single atoms of the Ce$_1$-Ru$_n$/NC catalyst during the alkaline HER and in turn made the fully-exposed Ru nanoclusters therein serve as hydrogen evolution centers, hence realizing the active site reverse for hydrogen evolution relative to the Ru$_1$-Ru$_n$ catalyst.

The Ce$_1$-Ru$_n$/NC and other control catalysts were then employed for electrochemical alkaline HER evaluations. As expected, the Ce$_1$-Ru$_n$/NC catalyst exhibited much enhanced catalytic activity than other control catalysts as indicated by their linear sweep voltammetry (LSV) curves in Fig. 5a. To make it clear, the current density of the Ce$_1$-Ru$_n$/NC catalyst at −0.05 V was up to 30 mA cm$^{-2}$, which was 2.3 times and 4.6 times that of the 20 wt% Pt/C and the Ru$_n$/NC catalyst as demonstrated in Fig. 5b. When the reaction potential reached −0.15 V, the current density of the Ce$_1$-Ru$_n$/NC catalyst was improved to 281 mA cm$^{-2}$ and became 3.5 times that of the 20 wt% Pt/C and 8.5 times that of the Ru$_n$/NC catalyst. Moreover, the mass activity of the Ce$_1$-Ru$_n$/NC catalyst was much larger than the 20 wt% Pt/C as shown in Fig. S25 (the calculation details were provided in methods part and the active metal loadings were shown in Table S3). We have also measured the Tafel slopes of these catalysts under steady-state conditions using the CA measurements as suggested previously[33,34]. It was demonstrated in Fig. 5c that

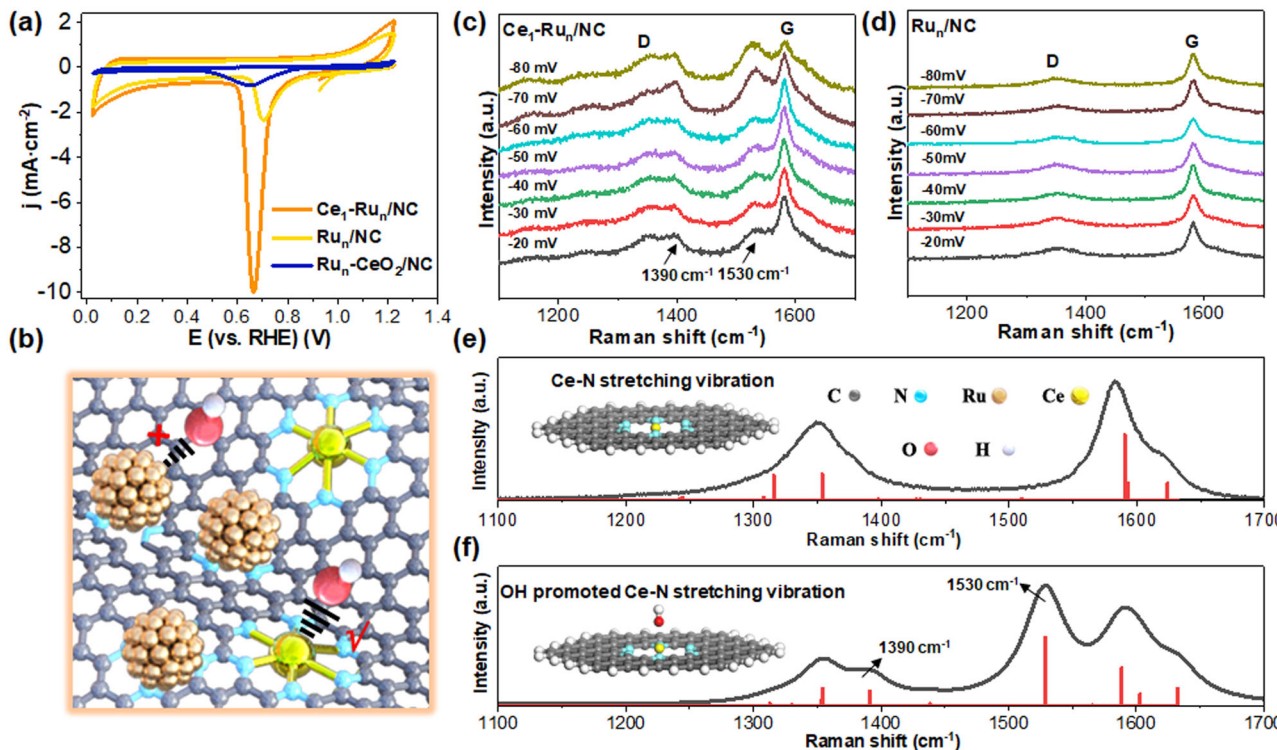

**Fig. 4 | Exploration of the interactions between OH and the Ce₁-Ruₙ/NC catalyst. a** CV curves of the Ce₁-Ruₙ/NC, Ruₙ/NC, and Ruₙ-CeO₂/NC catalysts measured in 1.0 M KOH solution. **b** Schematic illustration of the bonding trend between OH and Ce. In situ Raman spectra of the (**c**) Ce₁-Ruₙ/NC catalyst and the (**d**) Ruₙ/NC catalyst under real alkaline HER conditions. **e** Fitted Raman curve of the Ce-N stretching vibration in the blank Ce₁-Ruₙ/NC catalyst. **f** Fitted Raman curve of the Ce-N stretching vibration of the Ce₁-Ruₙ/NC in the presence of OH.

the obtained Tafel slope of the Ce₁-Ruₙ/NC catalyst (41.5 mV dec⁻¹) was markedly lower than that of the 20 wt% Pt/C catalyst (60.6 mV dec⁻¹), the Ruₙ/NC catalyst (134.4 mV dec⁻¹), and the Ruₙ-CeO₂/NC catalyst (180.7 mV dec⁻¹), indicating fast reaction kinetics for hydrogen evolution over the Ce₁-Ruₙ/NC catalyst. Additionally, the smallest charge transfer resistance ($R_{ct}$) of the Ce₁-Ru/NC among these catalysts as shown in Fig. S26 provided further evidence for its rapid HER kinetics. To examine the intrinsic activity of these catalysts, we have further conducted the turnover frequency (TOF) measurements for these catalysts in the potential range from −0.01 V to −0.06 V. It was found that the TOF values of the Ce₁-Ruₙ/NC catalyst and the Ruₙ/NC catalyst were obviously higher than the 20 wt%, Pt/C as shown in Fig. 5d. Moreover, with the assistance of Ce single atoms, the TOF value of the Ce₁-Ruₙ/NC catalyst was significantly larger than that of the Ruₙ/NC catalyst. However, the Ruₙ-CeO₂/NC catalyst showed obviously reduced TOF value relative to that of the Ce₁-Ruₙ/NC catalyst, which possibly derived from its weak OH adsorption strength as demonstrated by the aforementioned CV results in Fig. 4a.

To compare the alkaline HER activity of the Ce₁-Ruₙ/NC catalyst with the state-of-the-art Ru₁-Ruₙ/NC catalyst, we have also prepared a Ru₁-Ruₙ/NC catalyst by reducing the Ru³⁺ impregnated NC support at 180 ℃ with hydrogen (please see methods part for more details). The HAADF-STEM images and aberration-corrected HAADF-STEM images of the Ru₁-Ruₙ/NC catalyst in Fig. S27 unveiled the copresence of the uniformly dispersed Ru single atoms and Ruₙ nanoclusters (0.9 ± 0.2 nm) in the Ru₁-Ruₙ/NC catalyst. It was found that the alkaline HER activity of the Ru₁-Ruₙ/NC catalyst was markedly lower than that of the Ce₁-Ruₙ/NC catalyst as suggested both by their LSV curves and the TOF values of the two catalysts as displayed in Fig. S28. Moreover, in spite of the similar Ru loadings between the Ce₁-Ruₙ/NC catalyst and the Ru₁-Ruₙ/NC catalyst, the mass activity normalized to per milligram Ru of the Ce₁-Ruₙ/NC catalyst was markedly higher than that of the Ru₁-Ruₙ/NC catalyst as displayed in Fig. S29, which unveiled the higher Ru atom efficiency for alkaline hydrogen evolution over the Ce₁-Ruₙ/NC catalyst than that over the Ru₁-Ruₙ/NC catalyst.

To make an exact comparison of mass activity between the Ce₁-Ruₙ/NC catalyst and previously reported Ru-based alkaline HER catalysts, we have cited more than 100 papers published in recent years as summarized in Table S4-5 and Fig. 5e. It could be identified that the mass activities of the Ce₁-Ruₙ/NC catalyst measured at −0.1 V and −0.05 V were superior to most of the reported results. To examine the practical application potential of the Ce₁-Ruₙ/NC catalyst, we have further measured the durability of the Ce₁-Ruₙ/NC catalyst on an alkaline anion-exchange-membrane water electrolysis (AEMWE) device using the Ce₁-Ruₙ/NC as the cathodic catalyst and the nickel foam as the anodic catalyst. The chronopotentiometry curve acquired by setting the reaction temperature at 80 °C and the current density output at 400 mA/cm² in Fig. 5f suggested that the Ce₁-Ruₙ/NC catalyst displayed quite good stability for 100 h testing. We have also characterized the Ce₁-Ruₙ/NC catalyst after the stability test using a variety of techniques. To begin with, as displayed by the HAADF-STEM images of the spent Ce₁-Ruₙ/NC catalyst in Fig. S30a, b, uniformly dispersed Ruₙ nanoclusters with an average particle size of 1.0 ± 0.2 nm was identified, agreeing well with its initial particle size (1.0 ± 0.2 nm). Moreover, the aberration-corrected HAADF-STEM images of the post-reaction Ce₁-Ruₙ/NC catalyst in Fig. S30c, d unveiled the coexistence of the homogeneously distributed Ce single atoms and the Ruₙ nanoclusters on the NC support, which was also verified by corresponding EDS elementary mapping images in Fig. S30e–g. These characterization results demonstrated the robust stability of the Ce₁-Ruₙ/NC catalyst. In addition, we have also performed the continuous cyclic voltammogram measurement to examine the stability of the Ce₁-Ruₙ/NC catalyst. As shown in Fig. S31, after 6000 cyclic voltammogram cycles, the LSV curve of the Ce₁-Ruₙ/NC catalyst remained to be nearly overlapped

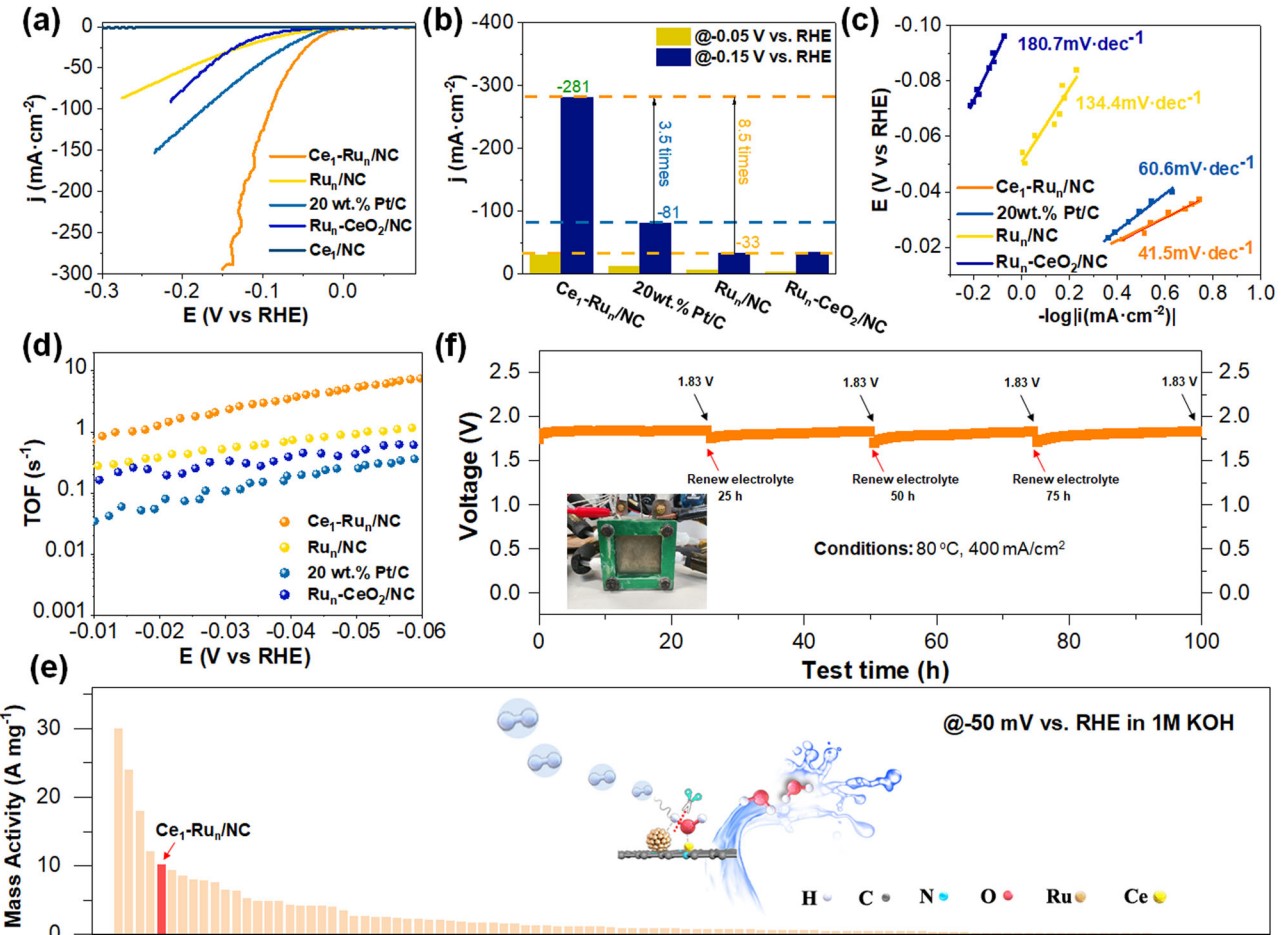

**Fig. 5 | Electrocatalytic alkaline HER evaluations of the Ce$_1$-Ru$_n$/NC and other control catalysts. a** LSV curves of the tested catalysts. **b** The comparison of current densities at −0.05 V and -0.15 V for the tested catalysts. **c** Tafel plots of the Ce$_1$-Ru$_n$/NC, Ru$_n$/NC, Ru$_n$-CeO$_2$/NC and the commercial 20 wt% Pt/C catalysts. **d** TOF value curves of the Ce$_1$-Ru$_n$/NC, Ru$_n$/NC, Ru$_n$-CeO$_2$/NC and the 20 wt% Pt/C catalysts in the potential range from −0.01 V to −0.06 V. **e** Comparison of the alkaline HER mass activity of the Ce$_1$-Ru$_n$/NC catalyst with previously reported Ru-based catalysts at −0.05 V vs. RHE. **f** The chronopotentiometry curve of the Ce$_1$-Ru$_n$/NC catalyst under AEMWE conditions at a reaction temperature of 80 °C and a current density of 400 mA/cm$^2$. The inset of (**f**) was the photograph of the AEMWE device.

with its initial one, which again disclosed the excellent durability of the Ce$_1$-Ru$_n$/NC catalyst.

## Theoretical investigations of the Ce$_1$-Ru$_n$/NC catalyst's superior alkaline HER activity

The first-principle density functional theory (DFT) calculations with spin-polarization were further carried out to explore the origin of the excellent alkaline HER activity of the Ce$_1$-Ru$_n$/NC catalyst. The structural models of the Ce$_1$-Ru$_n$/NC, Ru$_n$/NC and Ce$_1$/NC catalysts were built based on the EXAFS fitting data, wherein Ce single atom was coordinated with six N atoms and Ru single atom was stabilized by four N atoms of the NC support. In view of the copresence of Ru single atoms and Ru$_n$ nanoclusters in the Ru$_n$/NC catalyst, we have built a dual Ru$_1$-Ru$_{13}$ model for simulating the Ru$_n$/NC catalyst by considering both the particle size of the Ru$_n$ nanocluster (1 nm, Fig. S11) and its neighboring Ru single atoms (Ru$_1$) as was also commonly employed by previous reports for simulating the united catalyst with coexisted Ru single atoms and Ru nanoclusters/nanoparticle[23,35,36]. All the structure details of the constructed models were provided in supporting information (Figs. S32−35). To begin with, it should be noted that pure Ru single atoms on the NC support were insufficient for dissociating water molecule, which was regarded as the rate-determining step of alkaline HER. As demonstrated in Fig. S36, the Gibbs free energy barrier for water dissociation over the Ru single atom was as huge as 1.03 eV, indicating its poor water dissociation property and thus low alkaline

HER activity. To further examine the alkaline HER activity of the Ru single atoms, we have synthesized a Ru$_1$/NC catalyst (the synthetic details were demonstrated in methods part). The formation of Ru single atoms in the Ru$_1$/NC catalyst was confirmed by the aberration-corrected HAADF-STEM images of it in Fig. S37. As further revealed by the alkaline HER evaluation results (LSV curve) in Fig. S38, the catalytic activity of the Ru$_1$/NC catalyst was much lower than that of the Ru$_n$/NC catalyst. In the meanwhile, the dual Ru$_1$-Ru$_{13}$ sites presented a quite low Gibbs free energy barrier for dissociating water molecules compared with that of the pure Ru single atoms (Fig. S1 and Fig. S36), making the dual Ru$_1$-Ru$_{13}$ site a reasonable structural model for simulating the Ru$_n$/NC catalyst. As far as the Ce$_1$-Ru$_n$/NC catalyst was concerned, both the Ru single atoms and Ce single atoms in it were insufficient for dissociating water in terms of their huge Gibbs free energy barriers for water dissociation as demonstrated in Fig. S36 and Fig. S39, respectively. Keeping in mind that the experimental test results in Fig. 5 demonstrated that the alkaline HER activity of the Ce$_1$-Ru$_n$/NC catalyst was much higher than the Ru$_n$/NC catalyst, it was thus reasonable to use the dual Ce$_1$-Ru$_{13}$ model to simulate the Ce$_1$-Ru$_n$/NC catalyst because it was the sole structural difference between the Ce$_1$-Ru$_n$/NC catalyst and the Ru$_n$/NC catalyst.

Figure 6a shows that the Ce$_1$-Ru$_n$/NC catalyst possessed a micro-electric field due to the notable difference in the electrostatic potential distribution between the Ce single atoms and the fully-exposed Ru nanoclusters. Consequently, the water molecules could be easily

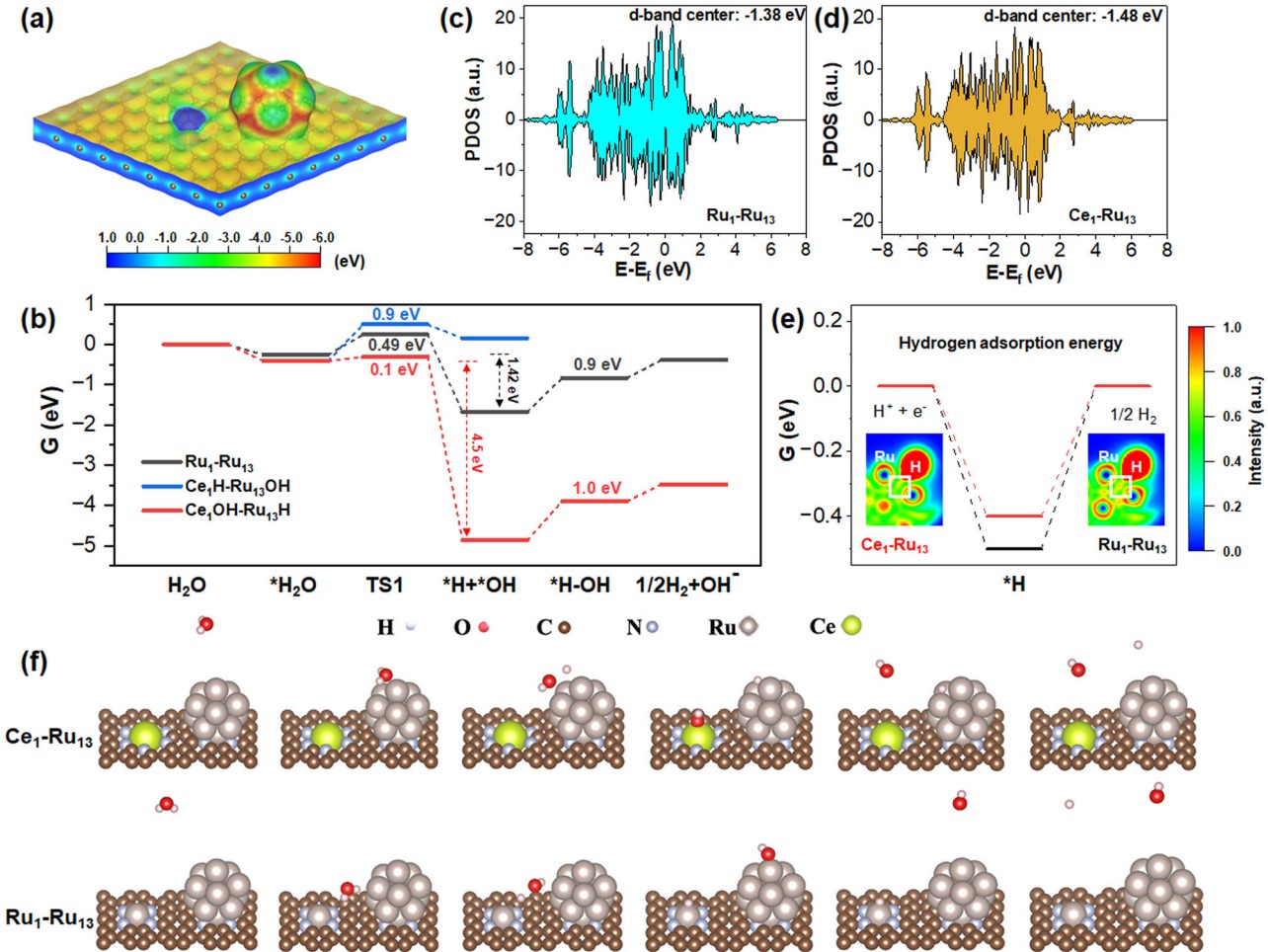

**Fig. 6 | Reaction energy diagrams of alkaline HER over the dual Ce$_1$-Ru$_{13}$ site and over the dual Ru$_1$-Ru$_{13}$ site. a** Surface electrostatic potential distribution of the dual Ce$_1$-Ru$_{13}$ site. **b** The Gibbs free energy diagrams for complete hydrogen evolution reaction over the dual Ce$_1$-Ru$_{13}$ site and the dual Ru$_1$-Ru$_{13}$ site. **c, d** Partial density of state (PDOS) of Ru$_{13}$ nanoclusters in the dual Ce$_1$-Ru$_{13}$ site and in the dual Ru$_1$-Ru$_{13}$ site, respectively. **e** The H adsorption energies on the dual Ce$_1$-Ru$_{13}$ site and on the dual Ru$_1$-Ru$_{13}$ site, respectively. The insets of (**e**) are the ELF pictures of *H on

the dual Ce$_1$-Ru$_{13}$ site and on the dual Ru$_1$-Ru$_{13}$ site, respectively. **f** The schematic illustrations for each step of the alkaline HER process over the dual Ce$_1$-Ru$_{13}$ site and over the dual Ru$_1$-Ru$_{13}$ site, respectively. The *H$_2$O indicated the adsorbed water molecules on the catalyst. The *H + *OH indicated the adsorbed H and the adsorbed OH on the catalyst, respectively. The *H·OH represented for that the H was still adsorbed on the catalyst while the OH was undergoing the process of desorption.

elongated upon contacting the Ce$_1$-Ru$_n$/NC catalyst. It was found that the O-H bond of water was significantly elongated from 0.98 Å to 1.04 Å when adsorbed on the Ce$_1$-Ru$_n$/NC catalyst, hence efficiently favoring water activation. In theory, water had two possible activation patterns on the Ce$_1$-Ru$_n$/NC catalyst: (i) the OH groups bonded with Ce single atoms while the H atoms bonded with Ru nanoclusters (Ce$_1$OH-Ru$_n$H route); and (ii) the H atoms bonded with Ce single atoms while the OH groups bonded with Ru nanoclusters (Ce$_1$H-Ru$_n$OH route). Therefore, we have performed the DFT calculations for both routes. As displayed in Fig. 6b, the Ce$_1$H-Ru$_n$OH route was endothermic by 0.6 eV while the Ce$_1$OH-Ru$_n$H route was particularly exothermic by 4.5 eV. The sharply different thermodynamics of the Ce$_1$H-Ru$_n$OH route and the Ce$_1$OH-Ru$_n$H route was most likely derived from the distinct oxophilicity of Ru and Ce. Compared with the d-block Ru, the f-block Ce was much more oxophilic[37] by means of which Ce was more prone to bond with OH instead of H, hence making the Ce$_1$OH-Ru$_n$H route more thermodynamically favored.

The DFT calculations further suggested that the exothermic energy of the dual Ce$_1$-Ru$_{13}$ site (4.5 eV) was 3.2 times that of the dual Ru$_1$-Ru$_{13}$ site (1.42 eV) in water activation process, which unveiled that the dual Ce$_1$-Ru$_{13}$ sites were much more thermodynamically favorable to dissociate water molecules than the dual Ru$_1$-Ru$_{13}$ sites. In addition,

the water dissociation energy barrier over the dual Ce$_1$-Ru$_{13}$ site (0.1 eV) was also markedly lower than that over the dual Ru$_1$-Ru$_{13}$ site (0.49 eV). Therefore, the dual Ce$_1$-Ru$_{13}$ sites were both thermodynamically and kinetically beneficial to dissociate water molecules relative to the dual Ru$_1$-Ru$_{13}$ sites. After water dissociation, OH was bonded with the oxophilic Ce single atoms while H was adsorbed on the fully exposed Ru nanoclusters.

To gain theoretical insight into hydrogen formation and OH desorption over the dual Ce$_1$-Ru$_{13}$ sites and Ru$_1$-Ru$_{13}$ sites, we have further carried out calculations for H binding strength ($\Delta G_{H*}$) and OH desorption energy barriers for both of the two models. Because of the electron enriching of Ru in the Ce$_1$-Ru$_n$/NC catalyst as revealed by the aforementioned XANES and XPS results, the calculated d-band center of the Ru$_n$ nanoclusters in the Ce$_1$-Ru$_n$/NC catalyst (−1.48 eV) displayed an obvious downshift compared with that of the Ru$_n$/NC catalyst (−1.38 eV) as shown in Fig. 6c, d. According to previous reports[30], the low reactivity of Ru metal toward HER was usually ascribed to its too strong H adsorption that hindered H transfer and H$_2$ formation. The d-band center downshift of the Ru$_n$ nanoclusters in the Ce$_1$-Ru$_n$/NC catalyst would facilely weaken H binding strength[38] to enable more feasible H transfer. As shown in Fig. 6e, the $\Delta G_{H*}$ of the dual Ce$_1$-Ru$_{13}$ site was obviously closer to the optimum value (0 eV) than the dual

$Ru_1$-$Ru_{13}$ site, which made the $Ce_1$-$Ru_n$/NC catalyst more efficient for hydrogen evolution than the $Ru_n$/NC catalyst. In another respect, the calculated H adsorption energies of the dual $Ce_1$-$Ru_{13}$ site and the dual $Ru_1$-$Ru_{13}$ site via the "computational hydrogen electrode model" method[39] were obviously larger than that of the experimentally measured onset potentials of them. This result was due to the fact that the reaction system was a grand canonical ensemble, which was significantly affected by the polarization effect. As a consequence, though the calculated reaction energy suggested the reasonable reaction trend, the value of the calculated reaction energy commonly displayed a marked deviation from the experimentally acquired result. Therefore, we have further performed the H adsorption energy calculations for the dual $Ce_1$-$Ru_{13}$ site and the dual $Ru_1$-$Ru_{13}$ site, respectively, taking the "CANDEL implicit solvation model" method. As indicated in Fig. S40, the calculated H adsorption energies on the dual $Ce_1$-$Ru_{13}$ site and on the dual $Ru_1$-$Ru_{13}$ site using this method were −0.096 eV and −0.142 eV, respectively, which were both closer to the experimentally measured onset potentials of them than that calculated through the "computational hydrogen electrode model" method. On the other hand, the electron localization function (ELF) analysis (inset of Fig. 6e) disclosed that a weaker degree of electron localization in the Ru-H bonding region was identified for the dual $Ce_1$-$Ru_{13}$ site than the dual $Ru_1$-$Ru_{13}$ site, which would largely weaken the hydrogen binding strength on the dual $Ce_1$-$Ru_{13}$ site for more favorable hydrogen formation.

In conclusion, we report that uniting highly oxophilic Ce single atoms and fully-exposed Ru nanoclusters on a N functionalized carbon support can efficiently reverse the alkaline hydrogen evolution centers to the more active $Ru_n$ nanocluster side, which largely boosted the alkaline HER activity of the catalyst. The facile active site reverse for hydrogen evolution was well confirmed both by the CV, in situ Raman measurements and the theoretical calculations. Notably, the mass activity of the $Ce_1$-$Ru_n$/NC catalyst (−10.1 A mg$^{-1}$ at −0.05 V) was superior to most of the reported Ru-based alkaline HER catalysts to date. In the meanwhile, the chronopotentiometry measurement performed on an AEMWE device further unveiled its excellent durability, which displayed great potential for practical applications. This finding possibly provides new insights into designing highly effective alkaline HER catalysts by rationally regulating the hydrogen evolution centers.

## Methods

### Materials
All chemicals were of analytical grade and used as received without further purification. Ruthenium chloride and urea were purchased from Macklin Biochemical Co., Ltd. Urea and carbon support (XC-72) were obtained from the Aladdin Biochemical Technology Co., Ltd. Cerium nitrate hexahydrate were purchased from the Sinopharm Chemical Reagent Co., Ltd. The 20% commercial Pt/C catalysts were purchased from the Sigma-Aldrich. The Nafion PFSA Polymer with 5% polymer content was acquired from the DuPont company. Ethanol was obtained from the Tongguang Fine Chemical Company Beijing. Hydrophilic carbon paper was purchased from the Kelude Experimental Equipment Technology Co., Ltd.

### Preparation of N functionalized carbon support (NC)
The N functionalized carbon supports were prepared according to our previous reports[40,41]. Typically, 2 g of XC-72 carbon was homogeneously grinded with 3 g of urea. After which, the mixture was calcined at 150 °C for 2 h and 300 °C for another 4 h in a fully sealed crucible. The product was then washed sufficiently with water and dried at 60 °C overnight.

### Synthesis of the $CeO_2$/NC
The preparation of $CeO_2$/NC was prepared via the impregnation method. To make it clear, 6 mg of $Ce(NO_3)_3 \cdot 6H_2O$ were firstly added to

20 mL of deionized water and stirred to dissolve the Ce precursor. Subsequently, 0.1 g of NC support were added to the above solution. After further stirring for 12 h, the mixture was centrifuged and washed thoroughly, and the obtained solid products were dried at 60 °C overnight. At last, the obtained powders were reduced with hydrogen at 700 °C for 2 h.

### Preparation of the $Ce_1$/NC
The prepared $CeO_2$/NC was then allowed for acid treatment (pH = 1) overnight. After sufficient washing and centrifugation, the obtained solid products were dried at 60 °C overnight to make the $Ce_1$/NC catalyst.

### Preparation details of the $Ce_1$-$Ru_n$/NC
Firstly, 2.5 mg of $RuCl_3$ were dissolved in 10 mL of deionized water. After which, 0.05 g of $Ce_1$/NC was added to the $RuCl_3$ solution under mild stirring. After stirring for another 12 h, the mixture was sufficiently washed, and the obtained solid products were collected and dried at 60 °C overnight. Finally, the obtained powders were reduced at 250 °C for 2 h under hydrogen atmosphere.

### Synthetic method of the $Ru_n$/NC
To begin with, 2.5 mg of $RuCl_3$ were added to 10 mL of deionized water and stirred to totally dissolve the Ru precursors. After which, 0.05 g of NC support were added to the $RuCl_3$ solution under mild stirring. After further stirring for 12 h, the mixture was sufficiently centrifugated and washed. The obtained solid powders were collected and dried at 60°C overnight. At last, the obtained powders were reduced with hydrogen at 250 °C for 2 h.

### Preparation of the $Ru_1$-$Ru_n$/NC
The preparation of the $Ru_1$-$Ru_n$/NC catalyst was similar to that of the synthesis of the $Ru_n$/NC catalyst. The only difference was that during the reduction of the $Ru_1$-$Ru_n$/NC catalyst, the reduction temperature was set at 180 °C.

### Preparation of the $Ru_1$/NC
The synthesis of $Ru_1$/NC catalyst was based on an impregnation method. To begin with, a small amount of $RuCl_3$ were dissolved in 10 mL of deionized water. Subsequently, 0.05 g of NC support were added to the solution and stirred for 12 h. After further centrifugation, the acquired solid powders were collected and dried at 60 °C overnight. Finally, the obtained powders were reduced with hydrogen at 180 °C for 2 h.

### Preparation of the $Ru_n$-$CeO_2$/NC
The preparation of the $Ru_n$-$CeO_2$/NC was as follows: 2.5 mg of $RuCl_3$ were firstly added to 10 mL of deionized water under mild stirring. Subsequently, 0.05 g of $CeO_2$/NC was added to the $RuCl_3$ solution under stirring. After stirring for another 12 h, the mixture was thoroughly centrifugated and washed. The acquired solid products were then collected and dried at 60 °C overnight. After which, the obtained powders were reduced with hydrogen at 250 °C for 2 h.

### Details of theoretical calculations
The slab structural models of the $Ce_1$-$Ru_n$/NC and the $Ru_n$/NC were constructed by taking consideration of the EXAFS fitting data in Table S1 to build the Ce-N and Ru-N coordination. A 20 Å-thick vacuum was introduced in the z-direction to avoid artificial interaction between periodic images. Periodic spin-polarized DFT calculations were conducted by virtue of the Vienna Ab-initio simulation package (VASP 5.4.4) with the Perdew-Burke-Ernzerhof (PBE) exchange-correlation function[42]. The core electrons were demonstrated through the projected augmented wave method[43]. Wherein, the valence electrons

(*Ce: 5s5p4f5d6s, Ru: 4p5s4d, O: 2s2p, N: 2s2p, C: 2s2p, H: 1s*) were expanded on plane-wave basis sets with a cutoff energy of 520 eV. A gamma point sampling based on the Monkhorst-Pack method was employed to calculate the slab structures. The self-consistent calculations applied a convergence energy threshold of $10^{-6}$ eV and a force threshold of 0.02 eV Å$^{-1}$. In another respect, the Van der Waals corrections with zero damping DFT-D3 method[44] of Grimme were taken in all structures. The Bader charge analysis was carefully carried out after each relaxation. The climbing image nudge elastic band (CI-NEB) method was conducted to demonstrate the energy barriers of the transition states during the $H_2O$ dissociation. The Gibbs free energy changes were acquired by the equation $\Delta G = \Delta E + \Delta ZPE - T\Delta S$, where $\Delta ZPE$ and $\Delta S$ were the changes in zero-point energy and entropy, as determined by taking into account of the adsorbed species and water molecules in the harmonic vibrational mode[45]. The climbing image nudged elastic band method[46] was employed to explore the TS and determine corresponding reaction barriers. The H adsorption energy was both calculated by the computational hydrogen electrode model method and the CANDEL implicit solvation model method. During calculations, the structures during the reaction were fully relaxed until the final force on each atom was less than 0.01 eV Å$^{-1}$. The H adsorption energies on the structure models of the dual $Ce_1$-$Ru_{13}$ site and the dual $Ru_1$-$Ru_{13}$ site were obtained by adding the vibrational contribution of H to the electronic energy of corresponding reaction systems. The Raman spectra were calculated through DFT calculations implemented by Gaussian 16 (A.03) program. The cluster model centered on a cerium atom was built based on the periodic model with the edge carbon atoms fixed and saturated by hydrogen atoms. Geometry optimization and frequency analysis were performed at the PBE0/(6-311 G* + SDD)[47] level of theory. Raman spectra were plotted by the Multiwfn package[48] with the frequency scaling factor of 0.988 and the full width at half maximum of peaks of 40 cm$^{-1}$.

## Details of characterizations

The crystallographic structures of materials were analyzed by the powder X-ray diffraction (XRD, Rigaku SmartLab, Cu Kα radiation, $\lambda = 1.54$ Å). The oxidation states of Ru were studied by the X-ray photoelectron spectroscopy (XPS, Thermo escalab 250XI Al Kα (hv = 1486.6 eV)). Transmission electron microscopy (TEM) analysis was performed on the FEI Talos F200X. The high angle annular dark field scanning TEM was operated at 200 kV. Atomic resolved STEM observation was carried out on ThemisZ (Titancubed·ThemisZ:300 kV·Thermo-Fisherscientific). The element amounts of samples were determined by the ICP-OES of Aligent ICPOES730. Both the Ru K-edge and Ce $L_{III}$-edge X-ray absorption spectra were collected at the XAS station (BL14W1) of the Shanghai Synchrotron Radiation Facility. The electron storage ring was carried out at 3.5 GeV with a maximum current of 250 mA. By a Si (311) double-crystal monochromator, the data was collected in fluoresce mode for both $L_{III}$-edge of Ce and K-edge of Ru. All the spectra data were recorded in room temperature. By utilization of a third ionization chamber, the standard of Ru foil, $RuO_2$, and $CeO_2$ were measured simultaneously with the sample for energy calibration purposes. The extended X-ray absorption fine structure (EXAFS) profiles for Ru K-edge and Ce $L_{III}$-edge were processed based on the standard procedures of the ATHENA module implemented in the IFEFFIT software packages. The $k^2$-weighted EXAFS spectra were acquired by subtracting the post-edge background from the overall absorption and subsequently normalizing with respect to the edge-jump step. The $k^2$-weighted $\chi(k)$ profiles of were Fourier transformed to real (R) space using a hanning windows (dk = 1.0 Å$^{-1}$) to separate the EXAFS contributions from different coordination shells. To acquire quantitative structural parameters around central atoms, least-squares curve parameter fitting was carried out using the ARTEMIS module of IFEFFIT software packages.

## Details of electrochemical measurements for alkaline HER

Electrochemical measurements for alkaline HER were carried out through an Autolab electrochemical station (Metrohm, Switzerland) in 1 M KOH solution at room temperature. The modules of electrochemical tests were managed by the computer software Nova 2. A typical three electrode system was used to study the electrochemical performance, in which the counter electrode and reference electrode were graphite rod and mercury/mercuric oxide electrode, respectively. All the potential value used in this work were calibrated to the reversible hydrogen electrode (RHE), using the following equation:

$$E_{RHE} = E_{Hg|HgO|KOH} + 0.098 + 0.059 \times pH$$

To prepare the catalyst ink, catalyst was homogeneously dispersed in 550 μl solution composed of 375 μl $H_2O$, 125 μl ethanol and 50 μl Nafion under sonication. 2.5 μl of the above-mentioned ink was then loaded on the glass carbon electrode with a diameter of 3 mm for alkaline HER tests. Alternatively, 20 μl of the catalyst ink was loaded onto 1 cm$^2$ hydrophilic carbon paper for stability testing. During all the alkaline HER evaluations, H-cell was used to eliminate the impacts of anode. The polarization curves were obtained by sweeping the potential from 0.1 to −0.325 V versus RHE at room temperature with a sweep rate of 9 mV s$^{-1}$ using the iR compensation. The stability of the $Ce_1$-$Ru_n$/NC was performed at 150 mA/cm$^{-2}$ via a chronopotentiometric technique.

The calculation of the turnover frequency (TOF, s$^{-1}$) for alkaline HER of the catalysts is shown below:

$$TOF = \frac{I_{product}/NF}{\omega \times m_{cat}/M_{Ru\,or\,Pt}} \times d$$

$I_{Product}$: Measured hydrogen currents, A;
N: Number of electron transfer (2 for $H_2$);
F: Faradaic constant, 96485 C mol$^{-1}$;
$m_{cat}$: Mass of catalyst on electrode, g;
$\omega$: Metal loading of the catalyst according to the ICP-OES results;
$M_{Ru}$: Atomic mass of Ru, 101.07 g mol$^{-1}$;
$M_{Pt}$: Atomic mass of Pt, 195.05 g mol$^{-1}$;
d: Mean particle size.
The calculation details for mass activity are also listed below:

$$I_{mass} = \frac{I}{\omega \times m_{cat}}$$

I: Measured currents, A;
$m_{cat}$: Mass of catalyst on electrode, g;
$\omega$: Metal loading of the catalyst based on the ICP-OES measurements;

## Details of in situ Raman measurements

Raman spectra were performed on an *inVia* Reflex instrument integrated with a confocal microscope. All spectra were acquired under room temperature with a 532 nm excitation. Calibration was performed with 520 cm$^{-1}$ peak (silicon wafer standard). To avoid damaging samples, the power of laser was decreased by a factor of 10. In situ Raman measurements were carried out on a home-made electrochemical reaction cell using a standard three electrode system in 1.0 M KOH electrolyte. Wherein, the Ag/AgCl electrode was used as the reference electrode, and the platinum wire served as the counter electrode. To prepare the working electrode, catalyst was firstly ultrasonically dispersed in 550 ml solvent (125 μl water, 375 μl ethanol and 50 μl Nafion solution). After which, 10 μl of the homogeneous ink was loaded onto a carbon paper with an area of 1 cm$^2$. The in situ Raman data were acquired from the chronoamperometry measurements performed in the potential range from 0 mV to −80 mV *versus*

RHE. The spectra were recorded at steady-state conditions by holding at the desired potential for at least 120 s.

## Data availability

All study data are included in the main text and the supporting information. Source data are provided as a Source Data file.

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

## Acknowledgements

This work was supported by the National Natural Science Foundation of China (22072153, 22102185), the Chinese Academy of Sciences and the Fundamental Research Funds for the Central Universities (E1E40310), National Key Research and Development Program of China (2022YFB3504200).

## Author contributions

K.Z. and M.H. conceived the project. F.S., Z.W., H.R., X.L., Z.C., and S.Y. prepared the catalysts and performed the characterizations. F.S., Z.W., H.R., G.S., and Y.C. conducted the catalytic measurements. Z.Z. carried out the DFT calculations and Z.H. helped with the experiment analysis. All authors discussed the results and co-wrote the paper.

## Competing interests

The authors declare no competing interests.
