## [Peer Review File · Nature Communications]

Oxophilic Ce single atoms-triggered active sites reverse for superior alkaline hydrogen evolutionREVIEWER COMMENTS

Reviewer #1 (Remarks to the Author):

This paper introduces a novel Ru catalyst design for alkaline HER reaction by including a single atom Ce onto the N functionalized carbon. Such a combination facilitates the high activity in alkaline HER reaction with outstanding catalytic stability. This work is significant to the catalyst field in water splitting by proposing a high-performance catalyst candidate that outperforms Pt-based catalysts.

The authors demonstrate that the Ce atom on the N-functionalized carbon can reverse the alkaline hydrogen evolution center so that the oxygen atom leans towards the more oxophilic Ce atom, compared to the Ru-based catalyst, where the oxygen atom is close to the Ru cluster. They also find a way to explain such mechanisms and compare different catalyst models using the computational simulation results. The methodology for the paper is logical, and all the discussions are reasonable.

In my opinion, this paper should be accepted.

Reviewer #2 (Remarks to the Author):

The authors presented that combining oxophilic Ce single atoms and fully exposed Ru nanoclusters leads to the outstanding alkaline HER activities by facilely reverse alkaline hydrogen evolution centers. They investigated the effects of oxophilic Ce and exposed Ru nanoclusters for facilitating water dissociation and alkaline hydrogen evolution. The authors provided the in-depth theoretical calculations and experimental results in this work. However, I think that this work lacks some important characterization analysis, and needs a major revision. The comments below should be addressed to be published on high-level Nature Communications.

1. In introduction part, the authors illustrate that the Ru₁-Ru_n catalyst can facilitate the water dissociation with a quite low Gibbs free energy barrier, in which the Ru single atoms (SAs) adsorbed the H while the small Ru nanoparticles (NPs) bonded with the OH, as illustrated in Scheme 1. However, they conflictingly illustrate that the Ru NPs of the Ru₁-Ru_n catalyst merely took part in water dissociation and just acted as spectators of hydrogen evolution, besides the Ru SAs were less active to produce hydrogen than the small Ru NPs. Although these conflict arguments have been unveiled in the cited references (Ref 18-22), the role of Ru SAs and Ru NPs can be tuned in various matrix for optimizing alkaline HER activity. The Ru NPs dominates the H₂ evolution compared to the Ru SAs on the metal oxide-based substrates such as sodium cobalt oxide that accelerates the water dissociation. [Adv. Mater.2023, 2301133] In contrast, on the carbon substrates such as defective carbon, the Ru SAs optimize the H₂ evolution while the Ru NPs facilitates the water dissociation. [Adv. Sci.2021,8, 20045] Given that the authors assert the reversed active centers to Ru nanoclusters by the oxophilic Ce SA as a main point of this study, the authors should make further detailed and clear explanations on these conflicting arguments.
2. To address the mentioned conflict arguments, please present a comparison between Ce₁-Ru_n/NC and Ru₁-Ru_n/NC by providing the experimental results and theoretical calculations including the HER performance and Gibbs free energy diagrams.
3. Some papers regarding to the alkaline HER are suggested into the introduction part after the revision process, including Nat. Mater. 2012, 11, 550–557, Mater. Today 2020, 6, 125-138, J. Am. Chem. Soc. 2016, 138, 16174-16181, J. Am. Chem. Soc. 2021, 143, 1399-1408,

Adv. Sci. 2018, 5, 1700464, J. Mater. Chem. A, 2019,7, 14971–15005.

4. Although the detailed synthetic procedures are illustrated in methods part and SI, it should be mentioned briefly in the main text (Results).

5. The figure 1a shows the schematic illustration of the structural models of the Ce1-Run/NC, but the author cited the figure 1a with explanations on the ICP-OES results. It should be corrected. The author should provide the results of ICP-OES as an additional table in SI.

6. The authors argued that the EDS mapping images indicated the uniform dispersion of the Ru nanoclusters and the Ce single atoms in Ce1-Run/NC. However, the EDS mapping images of Ce1-Run/NC (Figure 1r-u) were measured at lower magnification compared to those of the Ce1/NC and Run/NC, which shows no clear distribution of Ru nanoclusters and the Ce single atoms.

7. The Fourier transforms of the Ce L-edge EXAFS oscillations missed the description of the Ce-Ce coordination in Figure 2g.

8. The authors provide the Ru 3p XPS spectra. How about the XPS spectra for the other elements such as Ce, C and N? The full XPS results should be provided.

9. In 7 page, the author mentioned that Raman peak intensities were significantly strengthened as shown in Figure 3f, agreeing well with the in situ Raman spectra in Figure 3b. However, the Figure 3b indicates the schematic illustration of the bonding trend, not the in situ Raman spectra. Please check the figure notation, typo and spacing through the manuscript and SI.

10. The author exhibited the mass activities of the Ce1-Run/NC, Run/NC, Run-CeO₂/NC, and the commercial Pt/C catalysts, in Figure S21. The catalysts have different metal species. Please provide the mass loading for these catalysts.

11. The Ce1-Run/NC catalyst displayed excellent catalytic stability during chronopotentiometry at 150 mA cm⁻² for 100 hr. The author should provide the important characterization results of the catalyst after long-term HER process to reveal its stability. In addition, please provide the additional stability test of continuous cyclic voltammograms (> 1000 cycles).

12. In the Gibbs free energy diagrams (Figure 5b), what is the difference between *H+*OH and *H-OH steps? The authors might wish to provide the (schematic) illustrations for each step in the Gibbs free energy diagram.

Reviewer #3 (Remarks to the Author):

Shen et al. report on the synthesis of Ce single atoms and Ru nanoclusters on a N functionalized carbon and apply it for the alkaline HER. The claim of a record mass activity by the authors is not correct. Furthermore, the EXAFS and XANES data do not fit the assumed model of the catalyst. The DFT input model and DFT overpotentials do not align with the experimental analytic and catalytic data. Therefore, the manuscript in its current form is not suitable for publication in any journal.

Here are some of the main issues:

1. The authors cite the following manuscript in their main text: Li et al. J. Mater. Chem. A, 2021,9, 12196-12202. They also include a table to show the superior mass activity of their work compared to previous literature. Li et al.'s work is not shown in this Table (Table S1 and S2) and also not in Figure 4e of the main text. The authors show only works with a worse mass activity. However, Li et al. report a substantially better mass activity (17 A/g at 0.025 V overpotential). This publication was the only one I checked. There might be many

other works that reported better mass activities and that have been missed by the authors.

2. Is most of the Ru really in the form of metallic clusters? The XANES edge is way too high for metallic Ru. I cannot see how the XANES data would agree with the following statement: "It was demonstrated in Figure S8 that electrons could be facilely transferred from Ce to Ru with a net electron

transfer number of 0.05, which agreed well with the XANES data."

3. The authors claim that they have Ru clusters in their main compounds. In the DFT model, these clusters comprise more than 12 Ru atoms. This contradicts the EXAFS fitting, where Ru has, on average, only two neighbours. In a cluster of 12 or more, this number is tremendously larger. Thus, the majority of Ru is not present in the form of the assumed clusters.

4. The theoretical calculations do not align with the experimental observations. 1. The clusters are too large compared to the EXAFS coordination number. Figure 5e implies an overpotential of almost 400 mV, which does not fit to the experimentally observed one at all. Therefore, the DFT analysis is without any meaning for the real system.

5. The plot of the long-term stability measurement (Figure 4f) is missing ticks on the right y-axis to make it easy to read how much the sample deactivates. Using a ruler, I could find out that there is a substantial deactivation in the range of 50-100 mV. If the authors want to know if their material would be stable under industrial conditions, they must measure it at 80 °C and a current density of 400 mA/cm².

6. The general statement that the industry would prefer alkaline electrolysis is not true and is an oversimplification. It should be changed. Alkaline and PEM both have their advantages for industrial applications.

7. Replacing Platinum by Ruthenium is not a real advantage as they are similarly rare in the earth's crust.

8. Tafel slopes cannot be reliably determined from potentiodynamic methods. They must be measured under steady-state conditions with various CA measurements (see 10.1021/acseenergylett.1c00608 or /10.1016/j.mtener.2022.101123).

Response to the Reviewers' comments:

Reviewer 1

Overall comments: This paper introduces a novel Ru catalyst design for alkaline HER reaction by including a single atom Ce onto the N functionalized carbon. Such a combination facilitates the high activity in alkaline HER reaction with outstanding catalytic stability. This work is significant to the catalyst field in water splitting by proposing a high-performance catalyst candidate that outperforms Pt-based catalysts.

The authors demonstrate that the Ce atom on the N-functionalized carbon can reverse the alkaline hydrogen evolution center so that the oxygen atom leans towards the more oxophilic Ce atom, compared to the Ru-based catalyst, where the oxygen atom is close to the Ru cluster. They also find a way to explain such mechanisms and compare different catalyst models using the computational simulation results. The methodology for the paper is logical, and all the discussions are reasonable.

In my opinion, this paper should be accepted.

Response: We are very grateful for the Reviewer's highly positive assessment of our work.

Reviewer 2

Overall comments: *The authors presented that combining oxophilic Ce single atoms and fully exposed Ru nanoclusters leads to the outstanding alkaline HER activities by facilely reverse alkaline hydrogen evolution centers. They investigated the effects of oxophilic Ce and exposed Ru nanoclusters for facilitating water dissociation and alkaline hydrogen evolution. The authors provided the in-depth theoretical calculations and experimental results in this work. However, I think that this work lacks some important characterization analysis, and needs a major revision. The comments below should be addressed to be published on high-level Nature Communications.*

Response: We thank the Reviewer very much for reading through our manuscript and providing such valuable comments on our manuscript, which greatly helps us to improve our manuscript. In the revised manuscript and supporting information, we have updated the important characterizations as suggested by the Reviewer including the higher magnification EDS mapping images of the Ce₁-Ru_n/NC catalyst, the survey XPS spectra of the Ce₁-Ru_n/NC catalyst, the detailed characterizations for the spent Ce₁-Ru_n/NC catalyst after long-term HER measurement, and the CV stability test. Please see below our point-by-point responses to the Reviewer's concerns.

Comment 1: *In introduction part, the authors illustrate that the Ru₁-Ru_n catalyst can facilitate the water dissociation with a quite low Gibbs free energy barrier, in which the Ru single atoms (SAs) adsorbed the H while the small Ru nanoparticles (NPs) bonded with the OH, as illustrated in Scheme 1. However, they conflictingly illustrate that the Ru NPs of the Ru₁-Ru_n catalyst merely took part in water dissociation and just acted as spectators of hydrogen evolution, besides the Ru SAs were less active to produce hydrogen than the small Ru NPs. Although these conflict arguments have been unveiled in the cited references (Ref 18-22), the role of Ru SAs and Ru NPs can be tuned in various matrix for optimizing alkaline HER activity. The Ru NPs dominates the H₂ evolution compared to the Ru SAs on the metal oxide-based substrates such as sodium cobalt oxide that accelerates the water dissociation. [Adv. Mater.2023, 2301133] In contrast, on the carbon substrates such as defective carbon, the Ru SAs optimize the H₂ evolution while the Ru NPs facilitates the water dissociation. [Adv. Sci.2021,8, 20045] Given that the authors assert the reversed active centers to Ru nanoclusters by the oxophilic Ce SA as a main point of this study, the authors should make further detailed and clear explanations on these conflicting arguments.*

Response: Thanks a lot for the Reviewer's insightful comment. To better contextualize the work and highlight our idea of this study, we have included more detailed discussions in the revised manuscript. Firstly, there are some conflicting arguments regarding the hydrogen evolution mode over the Ru₁-Ru_n catalyst in previous reports, which was primarily derived from the influence of varied supports as was also pointed by the Reviewer. A general trend from a pure perspective of reaction energy was that Ru single atom itself was insufficient for dissociating water molecules (rate-determining step of alkaline HER) in terms of its huge energy barrier for water dissociation, due to lacking dual metal binding sites for simultaneously adsorbing H and OH group of water. By contrast, Ru nanoclusters with continuous Ru sites were more efficient for water dissociation with moderate energy barriers and meanwhile the Ru nanoclusters possessed more favorable H adsorption energy relative to that of the Ru single atoms. To better depict this point, we have provided more detailed DFT calculations.

As illustrated in Figure R1a, the Ru single atom (Ru₁) exhibited a particularly high Gibbs free energy barrier for water dissociation (up to 1.03 eV) while the value of the Ru_n nanocluster (n=13 in our case, corresponding to a Ru_n nanocluster of about 1 nm) was only 0.2 eV. In addition, water dissociation over the Ru_n nanocluster was the thermodynamically favorable exothermic process with an exothermic energy of 0.49 eV. Whereas, water dissociation was endothermic by 0.67 eV on the Ru₁ site. These results suggested that the Ru_n nanoclusters were much more efficient for dissociating water than the Ru single atoms. On the other hand, the H adsorption energy on the Ru_n nanocluster was -0.5 eV (Figure R1b), while the H adsorption energy on the Ru₁ site was -0.7 eV. These results revealed that the Ru_n nanoclusters were more

active for hydrogen evolution than the Ru single atoms both in terms of its favorable reaction energies for water dissociation and for H adsorption.

However, as far as the dual $\text{Ru}_1\text{-Ru}_n$ catalyst was concerned, the scenario became quite different. To begin with, water dissociation theoretically had two possible patterns on the dual $\text{Ru}_1\text{-Ru}_n$ site: i) the OH group bonded with Ru_1 while the H atom bonded with Ru_n ($\text{Ru}_1\text{OH-Ru}_n\text{H}$ route); and ii) the H atom bonded with Ru_1 while the OH group bonded with Ru_n ($\text{Ru}_1\text{H-Ru}_n\text{OH}$ route). To examine which route was the more energy-favored reaction pathway, we have further carried out the DFT calculations for both of the two routes. As displayed in Figure R2, the Gibbs free energy barrier for water dissociation via the $\text{Ru}_1\text{H-Ru}_n\text{OH}$ route was 0.49 eV, which was markedly lower than that via the $\text{Ru}_1\text{OH-Ru}_n\text{H}$ route (up to 0.7 eV), unveiling that the $\text{Ru}_1\text{H-Ru}_n\text{OH}$ route was the more kinetically favored reaction pathway for water dissociation. On the other hand, it was found that water dissociation via the $\text{Ru}_1\text{H-Ru}_n\text{OH}$ route was exothermic by 1.42 eV, while water dissociation via the $\text{Ru}_1\text{OH-Ru}_n\text{H}$ route was exothermic by only 0.03 eV, showing a 47-fold difference between them. This result further uncovered the more favorable thermodynamics for water dissociation via the $\text{Ru}_1\text{H-Ru}_n\text{OH}$ route relative to that via the $\text{Ru}_1\text{OH-Ru}_n\text{H}$ route. Therefore, driven by the more beneficial reaction kinetics and thermodynamics of the $\text{Ru}_1\text{H-Ru}_n\text{OH}$ route, H would be bonded with the Ru_1 and OH would be bonded with the Ru_n after water dissociation, which made further hydrogen evolution occur on the Ru_1 side of the dual $\text{Ru}_1\text{-Ru}_n$ site.

Figure R1. (a) The Gibbs free energy diagrams for water dissociation over the Ru_1 and over the Ru_n nanocluster ($n=13$), respectively. (b) The H adsorption energies on the Ru_1 and on the Ru_n nanocluster, respectively.

Figure R2. The Gibbs free energy diagrams for water dissociation via the $\text{Ru}_1\text{OH-Ru}_n\text{H}$ route and via the $\text{Ru}_1\text{H-Ru}_n\text{OH}$ route, respectively.

Whereas, this kind of hydrogen evolution mode via the $\text{Ru}_1\text{H-Ru}_n\text{OH}$ route had a major disadvantage that hydrogen evolution proceeded on the less reactive Ru_1 sites instead of the

more reactive Ru_n sites as described above in Figure R1a-b, which not only decreased the overall alkaline HER activity but unnecessarily required more usage of the precious Ru metal. To this end, new strategies are highly desired to reverse the hydrogen evolution centers from the Ru single atom side to the more reactive Ru_n nanocluster side of the Ru_1 - Ru_n catalyst to attain more efficient and cost-effective alkaline hydrogen evolution. Intriguingly, in current work, we found that by uniting the highly oxophilic Ce single atoms and the ultrafine Ru_n nanoclusters on a N functionalized carbon (NC) support, the hydrogen evolution sites were facily reversed to the Ru_n side due to the strong affinity of Ce to OH as verified both by the DFT calculations and the *in situ* Raman measurements in the main text, which greatly boosted the alkaline HER activity of the Ce_1 - Ru_n /NC catalyst. In another respect, it should be noted that the N functionalized XC-72 carbon support used in our work was highly graphitized in view of its low I_D/I_G value compared with the commercial XC-72 carbon of high graphitization degree as demonstrated in Figure R3, which largely excluded the influence of otherwise defective supports.

Figure R3. The Raman spectra of the Ce_1 - Ru_n /NC catalyst, Ru_n /NC catalyst, NC support, and the XC-72 carbon, respectively.

We have also added these contents to the revised manuscript as read in Line 18, Page 2, “From a perspective of reaction kinetics and thermodynamics, water dissociation over the dual Ru_1 - Ru_n site was more prone to proceed via a “ Ru_1H - Ru_nOH route” with OH of water adsorbed on the Ru_n side while the H bonded with the Ru_1 side as suggested by the theoretical calculations in Figure S1, making further hydrogen evolution primarily occur on the Ru_1 side of the dual Ru_1 - Ru_n catalyst. However, this kind of hydrogen evolution mode had a major disadvantage that the Ru_1 sites were less active for hydrogen evolution than the Ru_n sites both in view of the more favorable H adsorption energy and the lower energy barrier for water dissociation over the Ru_n sites than that over the Ru_1 sites as displayed in Figure S2, which led to significant decrease of the overall alkaline HER activity and also unnecessarily required more usage of the precious Ru metal. To this end, new strategies are highly desired to reverse the hydrogen evolution centers from the less reactive Ru single atom side to the more reactive Ru_n nanocluster side of the Ru_1 - Ru_n catalyst to attain more efficient and cost-effective alkaline hydrogen evolution, which is urgently awaited to be explored.”

Comment 2: To address the mentioned conflict arguments, please present a comparison between Ce_1 - Ru_n /NC and Ru_1 - Ru_n /NC by providing the experimental results and theoretical calculations including the HER performance and Gibbs free energy diagrams.

Response: Thanks for the Reviewer’s valuable comments and suggestions. As suggested by the Reviewer, we have prepared a Ru_1 - Ru_n /NC catalyst by reducing the Ru^{3+} impregnated NC

support at 180 °C with hydrogen (synthetic details were provided in methods part of the main text). As displayed by the HAADF-STEM image of the synthesized Ru₁-Ru_n/NC catalyst in Figure R4a-b, ultrafine Ru_n nanoclusters were uniformly dispersed on the NC support with an average particle size of 0.9±0.2 nm, which was similar to that of the Ru_n nanoclusters (1.0±0.2 nm) in the Ce₁-Ru_n/NC catalyst. Moreover, the aberration-corrected HAADF-STEM image of the Ru₁-Ru_n/NC catalyst in Figure R4c-d clearly showed the copresence of Ru single atoms and Ru_n nanoclusters in the Ru₁-Ru_n/NC catalyst. The loading amount of Ru in the Ru₁-Ru_n/NC catalyst was measured to be 1.1wt.% by the ICP-OES technique, which was also in line with that of the Ce₁-Ru_n/NC catalyst (1.1wt.%). Following the kind advice of the Reviewer, we have further evaluated the alkaline HER activity of the Ru₁-Ru_n/NC catalyst. As displayed in Figure R5a and b, the alkaline HER activity of the Ru₁-Ru_n/NC catalyst showed an obvious decrease relative to that of the Ce₁-Ru_n/NC catalyst both reflected by their linear sweep voltammetry (LSV) curves and corresponding turnover frequency (TOF) value curves of them.

Figure R4. (a) and (b) The HAADF-STEM images of the Ru₁-Ru_n/NC catalyst with varied magnifications. (c) and (d) The aberration-corrected HAADF-STEM images of the Ru₁-Ru_n/NC catalyst. The inset of (b) is the histogram of Ru_n particle-size distribution of the Ru₁-Ru_n/NC.

Figure R5. (a) The LSV curves for the Ru₁-Ru_n/NC catalyst and the Ce₁-Ru_n/NC catalyst during the alkaline HER evaluations. (b) The TOF value curves of the Ru₁-Ru_n/NC catalyst and the Ce₁-Ru_n/NC catalyst during the alkaline HER evaluations.

To gain theoretical insight into the catalytic activity difference between the Ce₁-Ru_n/NC catalyst and the Ru₁-Ru_n/NC catalyst, we have then conducted the first-principle DFT

calculations. As presented in Figure R6a, the water dissociation over the dual Ce₁-Ru₁₃ site (structural model of the Ce₁-Ru_n/NC catalyst) was highly exothermic by 4.5 eV, which was 3.2 times that of the dual Ru₁-Ru₁₃ site (structural model of the Ru₁-Ru_n/NC catalyst), suggesting the Ce₁-Ru_n/NC catalyst was much more thermodynamically favorable to dissociate water molecules than the Ru₁-Ru_n/NC catalyst. Moreover, the water dissociation energy barrier of the dual Ce₁-Ru₁₃ site (0.1 eV) was also markedly lower than that of the dual Ru₁-Ru₁₃ site (0.49 eV). These results unveiled both the beneficial reaction thermodynamics and kinetics of the Ce₁-Ru_n/NC catalyst for water dissociation than the Ru₁-Ru_n/NC catalyst. On the other hand, the H adsorption energy of the dual Ce₁-Ru₁₃ site was closer to the optimum value (0 eV) than the dual Ru₁-Ru₁₃ site as shown in Figure R6b. In the meanwhile, the electron localization function (ELF) analysis (inset of Figure R6b) disclosed the weaker degree of electron localization in the Ru-H bonding region for the dual Ce₁-Ru₁₃ site than that of the dual Ru₁-Ru₁₃ site. These results synergistically benefited the alkaline hydrogen evolution over the Ce₁-Ru_n/NC catalyst. It was also identified in Figure R6a that the OH desorption energy barriers of the dual Ru₁-Ru₁₃ site and the dual Ce₁-Ru₁₃ site were quite similar (0.9 eV vs. 1.0 eV), which showed a weak contribution to their alkaline HER activity difference.

Figure R6. (a) The Gibbs free energy diagrams for alkaline hydrogen evolution over the dual Ru₁-Ru₁₃ site and the dual Ce₁-Ru₁₃ site. (b) The H adsorption energies of the dual Ru₁-Ru₁₃ site and the dual Ce₁-Ru₁₃ site. The insets of (b) are the ELF pictures of *H on the dual Ce₁-Ru₁₃ site and on the dual Ru₁-Ru₁₃ site, respectively.

We have also updated these contents to the revised manuscript as read in Line 4, Page 10, “To compare the alkaline HER activity of the Ce₁-Ru_n/NC catalyst with the state-of-the-art Ru₁-Ru_n/NC catalyst, we have also prepared a Ru₁-Ru_n/NC catalyst by reducing the Ru³⁺ impregnated NC support at 180 °C with hydrogen (please see methods part for more details). The HAADF-STEM images and aberration-corrected HAADF-STEM images of the Ru₁-Ru_n/NC catalyst in Figure S27 unveiled the copresence of the uniformly dispersed Ru single atoms and Ru_n nanoclusters (0.9±0.2 nm) in the catalyst. It was found that the alkaline HER activity of the Ru₁-Ru_n/NC catalyst was markedly lower than that of the Ce₁-Ru_n/NC catalyst as suggested both by the LSV curves and the mass activity curves of the two catalysts in Figure S28.” Line 1, Page 13, “The DFT calculations suggested that the exothermic energy of the dual Ce₁-Ru₁₃ site (4.5 eV) was 3.2 times that of the dual Ru₁-Ru₁₃ site (1.42 eV) in water activation process, which unveiled that the dual Ce₁-Ru₁₃ sites were more thermodynamically favorable to dissociate water molecules than the dual Ru₁-Ru₁₃ sites. In addition, the water dissociation energy barrier of the dual Ce₁-Ru₁₃ site (0.1 eV) was also markedly lower than that of the dual Ru₁-Ru₁₃ site (0.49 eV). Therefore, the dual Ce₁-Ru₁₃ sites were both more thermodynamically and kinetically beneficial to promote water dissociation relative to the dual Ru₁-Ru₁₃ sites.” Line 22, Page 13, “As shown in Figure 5e, the ΔG_{H*} of the dual Ce₁-Ru₁₃ site was obviously closer to the optimum value (0 eV) than the dual Ru₁-Ru₁₃ site, which made the Ce₁-Ru_n/NC

catalyst more efficient for hydrogen evolution.” Line 39, Page 13, “Furthermore, the electron localization function (ELF) analysis (inset of Figure 5e) disclosed that a weaker degree of electron localization in the Ru-H bonding region was identified for the dual Ce₁-Ru₁₃ site than the dual Ru₁-Ru₁₃ site, thus weakening the hydrogen binding strength thereon for more favorable hydrogen production. It was also identified in Figure 5b that the OH desorption energy barriers of the dual Ru₁-Ru₁₃ site and the dual Ce₁-Ru₁₃ site were quite similar (0.9 eV vs. 1.0 eV), which showed a weak influence on their alkaline HER activity difference.”

Comment 3: *Some papers regarding to the alkaline HER are suggested into the introduction part after the revision process, including Nat. Mater. 2012, 11, 550–557, Mater. Today 2020, 6, 125-138, J. Am. Chem. Soc. 2016, 138, 16174-16181, J. Am. Chem. Soc. 2021, 143, 1399-1408, Adv. Sci. 2018, 5, 1700464, J. Mater. Chem. A, 2019,7, 14971–15005.*

Response: Thanks for the Reviewer’s kind reminding. We have cited these important references in the revised manuscript as was also listed below:

Ref 6: Adv Sci. 5, 1700464 (2018)

Ref 7: J Mater Chem A. 7, 14971-15005 (2019)

Ref 8: Nat Mater. 11, 550-557 (2012)

Ref 9: Mater Today. 36, 125-138 (2020)

Ref 10: J Am Chem Soc. 143, 1399-1408 (2021)

Ref 12: J Am Chem Soc. 138, 16174-16181 (2016)

Comment 4: *Although the detailed synthetic procedures are illustrated in methods part and SI, it should be mentioned briefly in the main text (Results).*

Response: Thanks for the Reviewer’s kind suggestion. In the revised manuscript, we have included the descriptions for the synthetic procedures in the main text as read in Line 17, Page 3, “In preparation, a N modification procedure was firstly conducted to functionalize the XC-72 carbon (denoted as NC support, please see methods part for details). Ce precursors were then introduced to the NC support by impregnation and allowed for further hydrogen reduction at 700 °C. After acid etching, the NC supported Ce single atoms were acquired (Ce₁/NC). To synthesize the united catalyst of oxophilic Ce single atoms and fully-exposed small Ru nanoclusters (Ce₁-Ru_n/NC), Ru precursors were then impregnated onto the Ce₁/NC and reduced by hydrogen at 250 °C. As a control, the Ru_n nanoclusters were also prepared on the NC support (denoted as Ru_n/NC) using a similar method to the synthesis of the Ce₁-Ru_n/NC catalyst except for the absence of Ce. The detailed synthetic process was further displayed in Figure S3 and in the methods part, respectively.”

Comment 5: *The figure 1a shows the schematic illustration of the structural models of the Ce₁-Ru_n/NC, but the author cited the figure 1a with explanations on the ICP-OES results. It should be corrected. The author should provide the results of ICP-OES as an additional table in SI.*

Response: Thanks a lot for the Reviewer for pointing out the misleading description. In the revised manuscript, we have corrected the sentence to “The schematic models of the Ce₁/NC, Ru_n/NC and Ce₁-Ru_n/NC were shown in Figure 1a, Figure 1h and Figure 1o, respectively.” In addition, as suggested by the Reviewer, we have also provided the metal loadings of the Ce₁/NC, Ru_n/NC, Ru₁-Ru_n/NC and Ce₁-Ru_n/NC catalysts as acquired by the ICP-OES measurement in Table R1 (also in Table S1 of the revised supporting information).

Table R1. The metal loading amount of the main catalysts in current work as measured by the

ICP-OES technique.

Catalyst	Ru loading (wt.%)	Ce loading (wt.%)
Ce ₁ /NC	-----	0.03
Ru _n /NC	1.2	-----
Ru ₁ -Ru _n /NC	1.1	-----
Ce ₁ -Ru _n /NC	1.1	0.03

The metal loading was determined by the inductively coupled plasma optical emission spectrometer (ICP-OES) measurement.

Comment 6: The authors argued that the EDS mapping images indicated the uniform dispersion of the Ru nanoclusters and the Ce single atoms in Ce₁-Ru_n/NC. However, the EDS mapping images of Ce₁-Ru_n/NC (Figure 1r-u) were measured at lower magnification compared to those of the Ce₁/NC and Ru_n/NC, which shows no clear distribution of Ru nanoclusters and the Ce single atoms.

Response: Motivated by the Reviewer's kind suggestion, we have further performed the energy-dispersive X-ray spectrometry (EDS) elementary mapping analysis of the Ce₁-Ru_n/NC catalyst under high magnifications. As displayed in Figure R7a and e, the aberration-corrected high-angle annular dark-field scanning transmission electron microscopy (HAADF-STEM) images of the Ce₁-Ru_n/NC catalyst disclosed uniformly dispersed single atoms and nanoclusters on the NC support. Corresponding EDS elementary mapping images of the Ce₁-Ru_n/NC catalyst in Figure R7b-d and Figure R7f-h suggested that the Ru element signal was concentrated on the small nanoclusters while the Ce element signal was highly overlapped with the distribution of single atoms. We have also modified the Figure R7a-d to the revised manuscript as displayed in the updated Figure 1r-u.

Figure R7. (a,e) The aberration-corrected HAADF-STEM images of the Ce₁-Ru_n/NC catalyst. (b)-(d) and (f)-(h) Corresponding energy-dispersive X-ray spectrometry (EDS) elementary mapping images of (a) and (e), respectively.

Comment 7: The Fourier transforms of the Ce L-edge EXAFS oscillations missed the description of the Ce-Ce coordination in Figure 2g.

Response: Thanks for the Reviewer's helpful comment. In the revised manuscript, we have included the discussion about the Ce-Ce coordination both for the Ce₁-Ru_n/NC catalyst and the reference CeO₂. As demonstrated by the Fourier transforms of the Ce L-edge EXAFS oscillations in Figure R8a (also in Figure 2g of the revised manuscript), a coordination peak in the second coordination shell at 3.8 Å was identified for the reference CeO₂. Corresponding data fitting of the reference CeO₂ EXAFS oscillation curve unveiled that this coordination peak was assigned to the Ce-Ce coordination. On the other hand, it was found that the Ce₁-Ru_n/NC

catalyst presented a coordination peak in the second coordination shell at 3.2 Å, which was shorter than that of the Ce-Ce coordination as displayed in Figure R8a. Careful data fitting showed that this peak was attributed to the Ce-C coordination of the Ce₁-Ru_n/NC catalyst in the second coordination shell. To further confirm this, we have carried out the high-resolution EXAFS wavelet transform (WT) characterizations both for the Ce₁-Ru_n/NC catalyst and the reference CeO₂. As shown in Figure R8b, the Ce₁-Ru_n/NC catalyst presented an intensity maximum at a K value of 2.3 Å⁻¹ that was derived from the Ce-C contribution in the second coordination shell. In sharp contrast, the Ce-Ce contribution in the reference CeO₂ induced an intensity maximum at a much higher K value of 7.7 Å⁻¹. This result again revealed the absence of Ce-Ce bond in the Ce₁-Ru_n/NC catalyst.

We have also added these contents to the revised manuscript as read in Line 32, Page 6, “We have also examined the Ce-Ce coordination in the Ce₁-Ru_n/NC catalyst and the reference CeO₂, respectively. As indicated by the Fourier transforms of the Ce L-edge EXAFS spectra in Figure 2g, a coordination peak at 3.8 Å was identified for the reference CeO₂, which was assigned to the Ce-Ce coordination by data fitting. Meanwhile, the Ce₁-Ru_n/NC catalyst presented a coordination peak at 3.2 Å that was markedly shorter than that of the Ce-Ce coordination. Careful data fitting suggested that this coordination peak of the Ce₁-Ru_n/NC catalyst was derived from the Ce-C contribution in the second coordination shell. To further confirm this, we have conducted the high-resolution EXAFS wavelet transform characterizations. As demonstrated in Figure S16, the reference CeO₂ displayed an intensity maximum at 7.7 Å⁻¹ (derived from the Ce-Ce contribution) while the Ce₁-Ru_n/NC catalyst showed an intensity maximum at 2.3 Å⁻¹ (originated from the Ce-C contribution). This result again uncovered the absence of Ce-Ce bond in the Ce₁-Ru_n/NC catalyst.”

Figure R8. (a) Fourier transforms of the Ce L_{III}-edge EXAFS oscillations in the R space of the Ce₁-Ru_n/NC catalyst and the reference CeO₂. (b) The wavelet transformation for the EXAFS signals of the Ce₁-Ru_n/NC catalyst and the reference CeO₂.

Comment 8: The authors provide the Ru 3p XPS spectra. How about the XPS spectra for the other elements such as Ce, C and N? The full XPS results should be provided.

Response: We thank the Reviewer for this helpful comment. As displayed in Figure R9a, we have provided the full XPS spectrum of the Ce₁-Ru_n/NC catalyst. It was found that the N 1s and C 1s XPS peaks of the Ce₁-Ru_n/NC catalyst were clearly identified both from the survey XPS spectrum and from the high-resolution N 1s and C 1s XPS spectra in Figure R9b and c, respectively. However, due to the low mass loadings of Ce in the Ce₁-Ru_n/NC catalyst, no Ce XPS peaks were found. Fortunately, the Ce L_{III}-edge XANES spectra in Figure R9d (also in Figure 2f) confirmed the presence of the Ce element in the Ce₁-Ru_n/NC catalyst thanks to the low detection limit of the synchrotron radiation technique.

We have also included these contents in the revised manuscript as read in Line 5, Page 7, “The full XPS spectrum of the Ce₁-Ru_n/NC catalyst was also provided in Figure S17, wherein the N 1s and C 1s XPS peaks were evidently identified. However, due to the low mass loadings

of Ce in the Ce₁-Ru_n/NC catalyst, no Ce XPS peaks were found in the survey XPS spectrum. Thanks to the low detection limit of the aforementioned XAFS measurement, the presence of Ce in the Ce₁-Ru_n/NC catalyst was well verified as demonstrated in Figure 2f.”

Figure R9. (a) Survey XPS spectrum, (b) high-resolution N 1s XPS spectrum, (c) high-resolution C 1s XPS spectrum of the Ce₁-Ru_n/NC catalyst. (d) Ce L_{III}-edge XANES spectra of the Ce₁-Ru_n/NC catalyst and the reference CeO₂.

Comment 9: In 7 page, the author mentioned that Raman peak intensities were significantly strengthened as shown in Figure 3f, agreeing well with the in situ Raman spectra in Figure 3b. However, the Figure 3b indicates the schematic illustration of the bonding trend, not the in situ Raman spectra. Please check the figure notation, typo and spacing through the manuscript and SI.

Response: Thanks for the Reviewer’s helpful comment. We sincerely apologize for the misleading typo and the Figure 3b in that sentence should be revised to **Figure 3c**. In the revised manuscript and supporting information, we have double-checked the figure notation, typo and spacing and made corrections accordingly.

Comment 10: The author exhibited the mass activities of the Ce₁-Ru_n/NC, Ru_n/NC, Ru_n-CeO₂/NC, and the commercial Pt/C catalysts, in Figure S21. The catalysts have different metal species. Please provide the mass loading for these catalysts.

Response: Thanks for the Reviewer’s valuable comment. As suggested by the Reviewer, we have provided the metal loadings of the tested catalysts in **Table R2** (also in **Table S3** of the revised supporting information). During the calculations, the mass activities of these catalysts were normalized to the mass of noble metals (Ru or Pt in current case) as commonly employed by previous reports. The calculation details were also provided below and in the revised supporting information.

$$I_{\text{mass}} = \frac{I}{\omega \times m_{\text{cat}}}$$

I: Measured currents, A;

m_{cat}: Mass of catalyst on electrode, g;

ω: Metal loading of catalyst based on the ICP-OES measurements;

Table R2. The loading amount of the Ru or Pt of the tested catalyst obtained by the ICP-OES measurement.

Catalyst	Ru loading (wt.%)	Pt loading (wt.%)
Ce ₁ -Ru _n /NC	1.1	-----
Ru _n /NC	1.2	-----
Ru _n -CeO ₂ /NC	1.3	-----
Commercial Pt/C	-----	20

The metal loading was determined by the inductively coupled plasma optical emission spectrometer (ICP-OES) measurement.

Comment 11: The Ce₁-Ru_n/NC catalyst displayed excellent catalytic stability during chronopotentiometry at 150 mA cm⁻² for 100 hr. The author should provide the important characterization results of the catalyst after long-term HER process to reveal its stability. In addition, please provide the additional stability test of continuous cyclic voltammograms (> 1000 cycles).

Response: As suggested by the Reviewer, we have employed a variety of techniques to characterize the Ce₁-Ru_n/NC catalyst after the stability test. Firstly, the high-angle annular dark-field scanning transmission electron microscopy (HAADF-STEM) images of the spent Ce₁-Ru_n/NC catalyst in Figure R10a-b showed the uniformly dispersed Ru_n nanoclusters with an average particle size of 1.0±0.2 nm in the catalyst, which was in good agreement with its initial particle size (1.0±0.2 nm). Other than that, the aberration-corrected HAADF-STEM observation of the post-reaction Ce₁-Ru_n/NC catalyst in Figure R10c-d further revealed the copresence of the homogeneously distributed Ce single atoms and the Ru_n nanoclusters on the NC support, which was also confirmed by corresponding EDS elementary mapping images in Figure R10e-g as the Ru element signal was primarily located on the small nanoclusters and the Ce element signal was overlapped with the dispersion of single atoms. These characterization results unveiled the robust stability of the Ce₁-Ru_n/NC catalyst.

In addition, we have further carried out the continuous cyclic voltammograms measurements for the Ce₁-Ru_n/NC catalyst following the Reviewer's kind suggestion. As shown in Figure R11, after 6000 cyclic voltammogram cycles, the obtained LSV curve of the Ce₁-Ru_n/NC catalyst remained to be nearly overlapped with its initial one, again indicating the excellent durability of the Ce₁-Ru_n/NC catalyst.

We have also included these contents in the revised manuscript as read in Line 24, Page 10, "We have also characterized the Ce₁-Ru_n/NC catalyst after the stability test using a variety of techniques. To begin with, as displayed by the HAADF-STEM images of the spent Ce₁-Ru_n/NC catalyst in Figure S29a-b, uniformly dispersed Ru_n nanoclusters with an average particle size of 1.0±0.2 nm was identified, agreeing well with its initial particle size (1.0±0.2 nm). Moreover, the aberration-corrected HAADF-STEM images of the post-reaction Ce₁-Ru_n/NC catalyst in Figure S29c-d unveiled the coexistence of the homogeneously distributed Ce single atoms and the Ru_n nanoclusters on the NC support, which was also verified by corresponding EDS elementary mapping images in Figure S29e-g. These characterization results demonstrated the robust stability of the Ce₁-Ru_n/NC catalyst. In addition, we have also performed the continuous cyclic voltammogram measurement to examine the stability of the Ce₁-Ru_n/NC catalyst. As shown in Figure S30, after 6000 cyclic voltammogram cycles, the LSV curve of the Ce₁-Ru_n/NC catalyst remained to be nearly overlapped with its initial curve, which again disclosed the excellent durability of the Ce₁-Ru_n/NC catalyst."

Figure R10. (a) and (b) The HAADF-STEM images of the post-reaction $\text{Ce}_1\text{-Ru}_n/\text{NC}$ catalyst. The inset of (b) is corresponding histogram of particle-size distribution of (b). (c) and (d): The aberration-corrected HAADF-STEM images of the post-reaction $\text{Ce}_1\text{-Ru}_n/\text{NC}$ catalyst. (e)-(g) Corresponding energy-dispersive X-ray spectrometry (EDS) elementary mapping images of (d).

Figure R11. The stability test of the $\text{Ce}_1\text{-Ru}_n/\text{NC}$ catalyst by virtue of the continuous cyclic voltammogram measurements conducted in 1.0 M KOH electrolyte.

Comment 12: In the Gibbs free energy diagrams (Figure 5b), what is the difference between $^\text{H}+^*\text{OH}$ and $^*\text{H-OH}$ steps? The authors might wish to provide the (schematic) illustrations for each step in the Gibbs free energy diagram.*

Response: Thanks for the Reviewer's valuable comments and suggestions. In Figure 5b, the $^*\text{H}+^*\text{OH}$ indicated the adsorbed H and the adsorbed OH on catalyst, respectively. The $^*\text{H-OH}$ represented for that the H was still adsorbed on the catalyst while the OH was undergoing the process of desorption. To make it clear, we have also added these descriptions to the revised manuscript, as read in Line 10, Page 11, "**The $^*\text{H}_2\text{O}$ indicated the adsorbed water molecules on**

the catalyst. The $*\text{H}+*\text{OH}$ indicated the adsorbed H and the adsorbed OH on the catalyst, respectively. The $*\text{H-OH}$ represented for that the H was still adsorbed on the catalyst while the OH was undergoing the process of desorption.”

To better show the reaction details for each step, we have provided the schematic illustrations for alkaline hydrogen evolution over the dual $\text{Ce}_1\text{-Ru}_{13}$ site and over the dual $\text{Ru}_1\text{-Ru}_{13}$ site, respectively, in Figure R12f below and also in the Figure 5f of the revised manuscript.

Figure R12. Reaction energy diagrams of alkaline HER over the dual $\text{Ce}_1\text{-Ru}_{13}$ site and over the dual $\text{Ru}_1\text{-Ru}_{13}$ site. (a) Surface electrostatic potential distribution of the dual $\text{Ce}_1\text{-Ru}_{13}$ site. (b) The Gibbs free energy diagrams for complete hydrogen evolution reaction over the dual $\text{Ce}_1\text{-Ru}_{13}$ site and over the dual $\text{Ru}_1\text{-Ru}_{13}$ site. (c,d) Partial density of state (PDOS) of Ru_{13} nanoclusters in the dual $\text{Ce}_1\text{-Ru}_{13}$ site and in the dual $\text{Ru}_1\text{-Ru}_{13}$ site, respectively. (e) The H adsorption energies on the dual $\text{Ce}_1\text{-Ru}_{13}$ site and on the dual $\text{Ru}_1\text{-Ru}_{13}$ site, respectively. The insets of (e) are the ELF pictures of $*\text{H}$ on the dual $\text{Ce}_1\text{-Ru}_{13}$ site and on the dual $\text{Ru}_1\text{-Ru}_{13}$ site, respectively. (f) The schematic illustrations for each step of the alkaline HER process over the dual $\text{Ce}_1\text{-Ru}_{13}$ site and over the dual $\text{Ru}_1\text{-Ru}_{13}$ site, respectively. The $*\text{H}_2\text{O}$ indicated the adsorbed water molecules on the catalyst. The $*\text{H}+*\text{OH}$ indicated the adsorbed H and the adsorbed OH on the catalyst, respectively. The $*\text{H-OH}$ represented for that the H was still adsorbed on the catalyst while the OH was undergoing the process of desorption.

Reviewer 3

Overall comments: Shen et al. report on the synthesis of Ce single atoms and Ru nanoclusters on a N functionalized carbon and apply it for the alkaline HER. The claim of a record mass activity by the authors is not correct. Furthermore, the EXAFS and XANES data do not fit the assumed model of the catalyst. The DFT input model and DFT overpotentials do not align with the experimental analytic and catalytic data. Therefore, the manuscript in its current form is not suitable for publication in any journal.

Response: We greatly appreciate the Reviewer's valuable comments, which helped us to improve the quality of our work. Also, we sincerely apologize for missing out the papers with higher mass activity than our work as mentioned by the Reviewer. In the revised manuscript, we have deleted the description of a record mass activity and cited more than 100 papers published in recent years to make an exact comparison of the mass activity between our work and previous reports. In addition, we have reconstructed the DFT models in the revised manuscript to better fit the EXAFS and XANES data. Other than that, we have provided detailed explanation on the difference between the overpotentials acquired by the theoretical calculations and the experiment measurements. Furthermore, we also tried to calculate the overpotential using a new "CANDEL implicit solvation model" method by means of which the calculated overpotential value was closer to that acquired by the experiment test. Please see below our point-by-point responses to the Reviewer's concerns.

Comment 1: The authors cite the following manuscript in their main text: Li et al. J. Mater. Chem. A, 2021,9, 12196-12202. They also include a table to show the superior mass activity of their work compared to previous literature. Li et al.'s work is not shown in this Table (Table S1 and S2) and also not in Figure 4e of the main text. The authors show only works with a worse mass activity. However, Li et al. report a substantially better mass activity (17 A/g at 0.025 V overpotential). This publication was the only one I checked. There might be many other works that reported better mass activities and that have been missed by the authors.

Response: Thanks for the Reviewer's kind reminding and we sincerely apologize for missing out the paper (J. Mater. Chem. A, 2021,9, 12196-12202) as pointed by the Reviewer. In the revised manuscript, we have deleted the claim of a record mass activity and cited more than 100 papers published in recent years hopefully to make an exact comparison of the mass activity between our Ce₁-Ru_n/NC catalyst and previous reports on Ru-based alkaline HER catalysts. As displayed in Table R3 and Table R4, there were indeed some better results in previous reports than our work in terms of mass activity. On the other hand, it was noteworthy that the mass activities of our Ce₁-Ru_n/NC catalyst measured at -0.1 V and -0.05 V (RHE) were superior to most of the reported Ru-based alkaline HER catalysts. We have also updated these contents in the revised manuscript as read in Line 13, Page 10, "To make an exact comparison of mass activity between the Ce₁-Ru_n/NC catalyst and previously reported Ru-based alkaline HER catalysts, we have cited more than 100 papers published in recent years as summarized in Table S4-5 and Figure 4e. It could be identified that the mass activities of the Ce₁-Ru_n/NC catalyst measured at -0.1 V and -0.05 V were superior to most of the reported results while there were some better results than the Ce₁-Ru_n/NC catalyst."

Table R3. The mass activity comparison between the Ce₁-Ru_n/NC catalyst and previously reported Ru-based alkaline HER catalysts at -0.05 V vs. RHE in 1M KOH electrolyte.

Catalysts	Ru loading (wt. %)	Mass activity /A mg _{Ru} ⁻¹	References
Ru/p-NC	0.7	ca. -30	J. Mater. Chem. A, 2021, 9, 12196-12202

Ru-NP@N-BP	0.44	-24	Catal. Sci. Technol., 2021, 11, 3182-3188
Ru-TA/ACC	0.058	ca. -18	J. Mater. Chem. A, 2019, 7, 11038-11043
Ru@Co-NC-800	0.35	-12	Appl. Surf. Sci. 2019, 494, 101-110
Ce₁-Ru_n/NC	1	-10.11	This work
Ru/WC	12.19	-9.33	J. Alloys Compd. 2023, 967, 171667
RuBi SAA/Bi@OG	NA	-8.5	Angew. Chem. Int. Ed. 2023, 62, e202300879
DR-Ru	0.8	-8	Sci China Mater 2021, 64(10): 2467-2476
Ru-HPC	5.55	-7.8	Nano Energy 2019, 58, 1-10
Ru/OMSNNC	1	-7.5	Adv. Mater. 2021, 33 (12), 2006965
Ru-ZIF-900	0.18	-6.4	J. Mater. Chem. A, 2020,8, 3203-3210
Ru/Na ⁺ , K ⁺ -PC	3.1	-6.3	Nano Res. 2023, 16 (7), 8836-8844
P-Ru/C	11.9	-5.2	ACS Catal. 2020, 10, 11751-11757
Ru@Cu-TM	NA	-4.87	Nat. Commun. 2023, 14 (1), 4680
RuCo@RuSACoSA-NMC	1.43	-4.85	Adv. Funct. Mater. 2023, 33 (40), 2301804
Ru NRs/TiN	2.26	-4.84	Appl. Catal., B 2022, 317, 121796
Ru/TiC	NA	-4.3	J. Electrochem. Sci. Technol., 2022, 13(4), 417
Ru/WNO@C	3.37	-4.2	Energy Environ. Sci., 2019, 12, 2569-2580
CoRu-O/A@HNC-2	1.28	-4.1	ACS Appl. Mater. Interfaces 2020, 12, 51437-51447
β -Ni(OH) ₂ /Ni-Ru SAs	1.17	-4.08	Adv. Funct. Mater. 2023, 2301343

Ru/Co@OG	6.9	-3.99	Angew.Chem. Int.Ed. 2021, 60,16044–16050
VO-Ru/HfO ₂ -OP	0.9	-3.5	Nat. Commun. 2022, 13 (1), 1270
Ru NPs/NC-900	10.5	-2.65	J. Am. Chem. Soc. 2022, 144, 19619–19626
RuCo@NC-750	1.56	-2.6	Electrochim. Acta 2019, 327, 134958
Ru ADC	4.25	-2.55	Small 2021, 17, 2101163
Ru _n -Ru _s /NC	4.01	-2.45	Small 2023, 2206949
RuGd-rGO	5.45	-2.3	Appl. Catal., B 2022, 304, 120916
Ru-CN/MC	5.21	-2.26	ChemElectroChem 2019, 6 (10), 2719-2725
Ru _{1,n} -ZnFe ₂ O _x -C	8.3	-2.21	Small 2022, 18, 2204155
Ru-NMCNs-500	3.04	-2.1	Int. J. Hydrogen Energy, 2020, 45, 18840e18849
MSOR ₁	4.09	-2.07	Adv. Funct. Mater. 2023, 33, 2210939
Ru@CN-0.16	3.18	-1.99	Energy Environ. Sci., 2018, 11, 800--806
NiCoRu _{0.2} /SP	0.9	-1.83	Appl. Catal., B., 2023, 331, 122710
Ru/V ₂ O ₃ -CC	0.37	-1.63	Int. J. Hydrogen Energy 2023, 48 (54), 20577-20587
Ru(3mL)-MC	2.65	-1.61	ChemElectroChem 2022, 9, e202101580
Ru@CQDs	3.78	-1.61	Adv. Mater., 2018, 30, 1800676
Ru/S-rGO	10.85	-1.58	ACS Appl. Mater. Interfaces 2020, 12, 43, 48591–48597
CoSA-NC@Ru	0.32	-1.5	Chem. Eng. J. 2022, 437, 135322
Ru/TiO ₂ -Vo@C-15	2.9	-1.29	J. Mater. Chem. A,2021, 9,10160–10168

Ru-MoS ₂ @PPy	6.05	-1.27	Int. J. Hydrogen Energy 2022, 47 (89), 37850-37859
Ru/NiFe LDH-F/NF	10	-1.24	Nanoscale, 2020, 12, 9669-9679
Ru-CON 800 °C	13.03	-1.2	Adv. Energy Mater. 2022, 12 (16), 2102257
CC@MoSe ₂ /Ru	0.78	-1.14	Adv. Mater. 2022, 34 (32), 2203900
Ru/NC-400	6	-1.1	Adv. Funct. Mater. 2021, 31, 2100698
Ru/TNTA	NA	-1.1	J. Power Sources 2023, 562, 232747
Ni@Ru/CNS	6.78	-0.95	Electrochimica Acta 2019, 320, 134568
RMC-500	7.13	-0.88	Int. J. Hydrogen Energy 2022, 47 (63), 26978-26986
HP-Ru/C	3.4	-0.86	Appl. Catal., B 2021, 294, 120230
Ru/MoO ₂	21.26	-0.85	Nano Energy 2021, 82, 105767
Ru/NC-0.01	16.7	-0.83	J. Mater. Chem. A, 2019, 7, 18072-18080
Ru-WSe ₂	4.25	-0.8	Inorg. Chem. Front., 2019, 6, 1382-1387
Ru/NSC-200	2.58	-0.8	J. Electroanal. Chem. 2023, 929, 117116
SL-RuO ₂ /C	21.66	-0.78	Int. J. Hydrogen Energy 2021, 46 (43), 22397-22408
RuNi NSs	NA	-0.74	Nano Energy 2019, 66, 104173
NC@Ru _{sa} -CoP	13.87	-0.74	Small 2023, 2301403
N-RuP/NPC	10.1	-0.73	Nano Energy 2020, 77, 105212
Ru/ZC	4	-0.71	J. Mater. Chem. A, 2022, 10, 17730-17739
Ru@F-Ni ₃ N	0.11	-0.7	ACS Appl. Mater. Interfaces 2022, 14, 32, 36688-36699

Ru/Ni/WC@NPC	4.13	-0.6	Adv. Energy Mater. 2022, 12, 2200332
Ru/CoO	NA	-0.6	J. Energy Chem. 2019, 37, 143-147
Ru@NCN	9.1	-0.58	J. Mater. Chem. A, 2021, 9, 13958–13966
S-RuP ₂ /NPC	4.8	-0.58	Int. J. Hydrogen Energy 2019, 44 (47), 25632-25641
Ru MIs-MoS ₂	10	-0.55	Chem. Eng. J. 2023, 453, 139803
3.0 wt% Ru/rGO	3	-0.5	Int. J. Hydrogen Energy 2022, 47 (94), 39853-39863
G(Zn)-Ru(100:1)	13.5	-0.46	ACS Appl. Mater. Interfaces 2021, 13, 28, 32997–33005
Ru/B–Ni ₂ P/Ni ₅ P ₄	6.28	-0.43	J. Mater. Chem. A, 2022, 10, 16236-16242
RuO _x @TiO ₂	10	-0.43	Chem. Eng. J., 2022, 446, 137248
Ru/H–S,N–C	17.1	-0.42	Nanoscale Adv., 2021, 3, 5068-5074
Ru/ α -MoC	7.65	-0.42	Appl. Catal., B 2022, 318, 121867
CoRu _{0.05} @Gr	2.48	-0.42	Appl. Surf. Sci., 2022, 580, 152294
Ru@RuP/PC-2	14.9	-0.41	J. Mater. Chem. A, 2019, 7, 5621-5625
Ru/AgCl@Ag-20%	NA	ca. -0.4	J. Phys. Chem. Lett. 2020, 11, 9, 3436– 3442
CNT-V-Fe-Ru	17.84	-0.4	ACS Catal. 2023, 13, 49–59
Ru-N/DOMMC	2.6	-0.31	https://doi.org/10.1002/cjoc.202300414
Co@RuCo-3	4.31	-0.3	Appl. Catal., B 2022, 315, 121554
RuPx@NPC	12.64	-0.3	ChemSusChem 2018, 11, 743 –752
Ru-CoO@SNG	ca. 9.9	-0.29	J. Electroanal. Chem. 2023, 932, 117272
Ru-HCRLs850	23.5	-0.22	J. Mater. Chem. A, 2019, 7, 6676-6685

RuCo@NG/N-GNs	64	-0.21	Chem. Eng. J. 2021, 422, 130077
Ru _{NP} -Ru _{SA} @CFN-800	12	-0.21	Adv. Funct. Mater. 2023, 2213058
RuO ₂ @MoS ₂	NA	-0.21	Appl. Surf. Sci. 2021, 538, 148019
Ru-HMT-MP-7	0.27	-0.2	Small 2022, 18 (11), 2105168
Ru-Cl-N SAC	24.3	-0.2	Chem. Eng. J. 2022, 441, 136078
RuNi-0	NA	-0.2	Int. J. Hydrogen Energy 2022, 47 (73), 31330-31341
Ru@SC-CDs2:10	21.6	-0.2	Nano Energy 2019, 65 104023
np-Cu ₅₃ Ru ₄₇	56.53	-0.19	ACS Energy Lett. 2020, 5 (1), 192-199
Ru-NiCo-LDH	NA	-0.19	Electrochem. Commun. 2019, 101, 23-27
Ru/Co ₄ N-CoF ₂	5	-0.17	Chem. Eng. J. 2021, 414, 128865
RuAu-0.2	74	-0.16	Adv. Energy Mater. 2019, 9 (20), 1803913
RuSe ₂	39	-0.15	Small 2021, 17 (13), 2007333
RuP ₂ @PC	14.6	-0.14	J. Mater. Chem. A, 2021, 9, 12276-12282
Ru/3DNCN	29	ca. -0.14	Appl. Catal., B 2021, 280, 119412
Ni ₅ P ₂ -mPtRu/NF	18.6	-0.12	Nanotechnology 2019, 30 (48), 485403
Ru ₁ Ni ₁ -NCNFs	28.2	-0.12	Adv. Sci. 2020, 7, 1901833
Sr ₂ RuO ₄	29.6	-0.1	Nat. Commun. 2019, 10 (1), 149
Ru@ZIF-L(Co)/FL-Ti ₃ C ₂ T _x	~25	ca. -0.1	Int. J. Hydrogen Energy 2022, 47 (77), 32787-32795
Ru-Cr ₂ O ₃ /NG	17	-0.08	RSC Adv., 2021, 11, 6107-6113
Ru _{1.0} /NF	1.1	ca. -0.07	ChemSusChem 2019, 12 (12), 2780-2787
CoMoRu _{0.25} O _x /NF	1.86	-0.04	Int. J. Hydrogen Energy 2022, 47 (94),

Cu _{1.94} S@Ru	55	-0.04	Small 2017, 13, 1700052
RuV–NiCoP/NF	NA	-0.03	J. Mater. Chem. A, 2021, 9, 26852-26860
CF@Ru–CoCH	NA	-0.03	Electrochimica Acta 2020, 331, 135367
Ru(OH) _x /Ag	3.27	-0.01	Int. J. Hydrogen Energy 2019, 44 (39), 21683-21691
W-Ru/NiP ₂	NA	-0.01	Appl. Surf. Sci. 2020, 508, 145302

Table R4. The mass activity comparison between the Ce₁-Ru_n/NC catalyst and previously reported Ru-based alkaline HER catalysts at -0.1 V vs. RHE in 1M KOH electrolyte.

Catalysts	Ru loading (wt.%)	Mass activity /A mg _{Ru} ⁻¹	References
Ru-NP@N-BP	0.44	ca. -50	Catal. Sci. Technol., 2021, 11, 3182-3188
Ru-TA/ACC	0.058	ca. -45	J. Mater. Chem. A, 2019, 7, 11038-11043
Ru/p-NC	0.7	NA	J. Mater. Chem. A, 2021, 9, 12196-12202
Ru@Co-NC-800	0.35	NA	Appl. Surf. Sci. 2019, 494, 101-110
Ce₁-Ru_n/NC	1	-44.3	This work
DR-Ru	0.8	-40	Sci China Mater 2021, 64(10): 2467–2476
RuBi SAA/Bi@OG	NA	-27.3	Angew. Chem. Int. Ed. 2023, 62, e202300879
Ru/WC	12.19	-24.83	J. Alloys Compd. 2023, 967, 171667
β-Ni(OH) ₂ /Ni-Ru SAs	1.17	-23.9	Adv. Funct. Mater. 2023, 2301343
Ru@WNO-C	0.9	-22.7	Nano Energy 2021, 80, 105531

PtCoRu NAs	0.85	-21.1	J. Colloid Interface Sci. 2020, 561, 372-378
Ru-ZIF-900	0.18	-19.8	J. Mater. Chem. A, 2020,8, 3203-3210
Ru-HPC	5.55	-17.9	Nano Energy 2019, 58, 1-10
Ru/Na ⁺ , K ⁺ -PC	3.1	-15.7	Nano Res. 2023, 16 (7), 8836-8844
RuCo@RuSACoSA-NMC	1.43	-14.9	Adv. Funct. Mater. 2023, 33 (40), 2301804
Ru ₁ /D-NiFe LDH	1.2	-14.7	Nat. Commun. 2021, 12 (1), 4587
Ru/OMSNNC	1	-14.7	Adv. Mater. 2021, 33 (12), 2006965
P-Ru/C	11.9	-12.9	ACS Catal. 2020, 10, 11751–11757
VO-Ru/HfO ₂ -OP	0.9	-12	Nat. Commun. 2022, 13 (1), 1270
CoRu-O/A@HNC-2	1.28	-10.1	ACS Appl. Mater. Interfaces 2020, 12, 51437–51447
Ru/TiC	NA	-9.9	J. Electrochem. Sci. Technol., 2022, 13(4), 417-423
Ru _n -Ru _s /NC	4.01	-8.8	Small 2023, 2206949
CC@MoSe ₂ /Ru	0.78	-7.81	Adv. Mater. 2022, 34 (32), 2203900
Ru/TNTA	NA	-7.1	J. Power Sources 2023, 562, 232747
NiCoRu _{0.2} /SP	0.9	-7	Appl. Catal., B., 2023, 331, 122710
RuGd-rGO	5.45	-5.98	Appl. Catal., B 2022, 304, 120916
Ru NPs/NC-900	10.5	-5.8	J. Am. Chem. Soc. 2022, 144, 19619–19626
RuCo@NC-750	1.56	-5.8	Electrochim. Acta 2019, 327, 134958
Ru-NMCMs-500	3.04	-5.2	Int. J. Hydrogen Energy. 2020, 45, 18840-18849

Ru ADC	4.25	-4.89	Small 2021, 17, 2101163
Ru/NiFe LDH-F/NF	10	-4.79	Nanoscale,2020, 12, 9669–9679
S-RuP@NPSC-900	0.8	-4.7	Adv. Sci. 2020, 7 (17), 2001526
Ru@CN-0.16	3.18	-4.52	Energy Environ. Sci., 2018, 11, 800–806
Ru/S-rGO	10.85	-3.87	ACS Appl. Mater. Interfaces 2020, 12, 43, 48591–48597
(Ru) RP-CPM	13.7	-3.85	Sci. Adv. 2020, 6 (44), eabb4197
Ru/V ₂ O ₃ -CC	0.37	-3.49	Int. J. Hydrogen Energy 2023, 48 (54), 20577-20587
Ru/NC-400	6	-3.4	Adv. Funct. Mater. 2021, 31, 2100698
Ru/TiO ₂ -Vo@C-15	2.9	-3.17	J. Mater. Chem. A,2021, 9,10160–10168
NCPO-Ru NCs	3.47	-3.02	ACS Nano 2022, 16, 5, 7993–8004
Ru–MoS ₂ @PPy	6.05	-2.87	Int. J. Hydrogen Energy 2022, 47 (89), 37850-37859
Ru@CQDs	3.78	-2.82	Adv. Mater., 2018, 30, 1800676
Ru/Mo ₂ CT _x	1.2	-2.8	Adv. Funct. Mater. 2023, 33 (16), 2214375
N-RuP/NPC	10.1	-2.72	Nano Energy 2020, 77, 105212
Ru NCs/VN-C ₃ N ₄	5.57	-2.69	J. Mater. Chem. A, 2023, 11, 18375-18386
RuP _x @NPC	12.64	-2.68	ChemSusChem 2018, 11,743 –752
Ru@F–Ni ₃ N	0.11	-2.46	ACS Appl. Mater. Interfaces 2022, 14, 32, 36688–36699
Ni@Ru/CNS	6.78	-2.42	Electrochimica Acta 2019, 320, 134568
Ir–Ru DSACs	NA	-2.36	Angew. Chem., Int. Ed. 2023, 62 (26), e202300873

HP-Ru/C	3.4	-2.3	Appl. Catal., B 2021, 294, 120230
Ru/NSC-200	2.58	-2.22	J. Electroanal. Chem. 2023, 929, 117116
NC@Ru _{sa} -CoP	13.87	-2.21	Small 2023, 2301403
Ru/CoO	NA	-2.19	J. Energy Chem. 2019, 37, 143-147
Ru/NC-0.01	16.7	-2.15	J. Mater. Chem. A, 2019, 7, 18072-18080
CNT-V-Fe-Ru	17.84	-2.07	ACS Catal. 2023, 13, 49-59
Ru-WSe ₂	4.25	-1.9	Inorg. Chem. Front., 2019, 6, 1382-1387
RMC-500	7.13	-1.89	Int. J. Hydrogen Energy 2022, 47 (63), 26978-26986
Ru/MoO ₂	21.26	-1.88	Nano Energy 2021, 82, 105767
Ru/ZC	4	-1.88	J. Mater. Chem. A, 2022, 10, 17730-17739
Ru@NCN	9.1	-1.83	J. Mater. Chem. A, 2021, 9, 13958-13966
Co@RuCo-3	4.31	-1.82	Appl. Catal., B 2022, 315, 121554
RuO _x @TiO ₂	10	-1.81	Chem. Eng. J., 2022, 446 137248
PdO-RuO ₂ /C	10.7	-1.7	ChemSusChem 2021, 14 (9), 2112-2125
S-RuP ₂ /NPC	4.8	-1.63	Int. J. Hydrogen Energy 2019, 44 (47), 25632-25641
3.0 wt% Ru/rGO	3	-1.46	Int. J. Hydrogen Energy 2022, 47 (94), 39853-39863
Ru/B-Ni ₂ P/Ni ₅ P ₄	6.28	-1.46	J. Mater. Chem. A, 2022, 10, 16236-16242
Ru MIS-MoS ₂	10	-1.45	Chem. Eng. J., 2023, 453 139803
G(Zn)-Ru(100:1)	13.5	-1.43	ACS Appl. Mater. Interfaces 2021, 13, 28, 32997-33005
Ru/ α -MoC	7.65	-1.42	Appl. Catal., B 2022, 318, 121867

CoRu _{0.05} @Gr	2.48	-1.35	Appl. Surf. Sci., 2022, 580, 152294
Ru/Ni/WC@NPC	4.13	-1.29	Adv. Energy Mater. 2022, 12, 2200332
Ru/WNO@C	3.37	-1.22	Energy Environ. Sci., 2019, 12, 2569-2580
Ru@RuP/PC-2	14.9	-1.01	J. Mater. Chem. A, 2019, 7, 5621-5625
Ru-CoO@SNG	ca. 9.9	-0.94	J. Electroanal. Chem. 2023, 932, 117272
RuP ₂ @PC	14.6	-0.93	J. Mater. Chem. A, 2021, 9, 12276–12282
Ru-N/DOMMC	2.6	-0.85	https://doi.org/10.1002/cjoc.202300414
RuSe ₂	39	-0.82	Small 2021, 17 (13), 2007333
Ru-HMT-MP-7	0.27	-0.79	Small 2022, 18 (11), 2105168
RuNi-0	NA	-0.62	Int. J. Hydrogen Energy 2022, 47 (73), 31330-31341
RuO ₂ @MoS ₂	NA	-0.52	Appl. Surf. Sci. 2021, 538, 148019
Ni ₅ P ₂ -mPtRu/NF	18.6	-0.51	Nanotechnology 2019, 30 (48), 485403
Ru-Cl-N SAC	24.3	-0.5	Chem. Eng. J. 2022, 441, 136078
np-Cu ₅₃ Ru ₄₇	56.53	-0.4	ACS Energy Lett. 2020, 5 (1), 192-199
Ru@SC-CDs2:10	21.6	-0.39	Nano Energy 2019, 65, 104023
RuCo@NG/N-GNs	64	-0.39	Chem. Eng. J. 2021, 422, 130077
Sr ₂ RuO ₄	29.6	-0.37	Nat. Commun. 2019, 10 (1), 149
CoMoRu _{0.25} O _x /NF	1.86	-0.37	Int. J. Hydrogen Energy 2022, 47 (94), 39908-39916
Ru-Cr ₂ O ₃ /NG	17	-0.36	RSC Adv., 2021, 11, 6107–6113
Ru/N,S- Go	7.2	-0.35	Appl. Surf. Sci. 2021, 551, 148742
Ru/3DNCN	29	ca. -0.31	Appl. Catal., B 2021, 280, 119412

Cu-RuS ₂ /Ru	30.2	-0.29	Mater. Today Phys. 2022, 23, 100625
Ru@ZIF-L(Co)/FL-Ti ₃ C ₂ T _x	~25	ca. -0.28	Int. J. Hydrogen Energy 2022, 47 (77), 32787-32795
Ru/Co ₄ N-CoF ₂	5	-0.28	Chem. Eng. J., 2021, 414, 128865
CF@Ru-CoCH	NA	-0.16	Electrochimica Acta 2020, 331, 135367
Cu _{1.94} S@Ru	55	-0.11	Small 2017, 13, 1700052
Ru(OH) _x /Ag	3.27	-0.05	Int. J. Hydrogen Energy 2019, 44 (39), 21683-21691
W-Ru/NiP ₂	NA	-0.02	Appl. Surf. Sci. 2020, 508, 145302

Comment 2: Is most of the Ru really in the form of metallic clusters? The XANES edge is way too high for metallic Ru. I cannot see how the XANES data would agree with the following statement: “It was demonstrated in Figure S8 that electrons could be facily transferred from Ce to Ru with a net electron transfer number of 0.05, which agreed well with the XANES data.”.

Response: We thank the Reviewer for these very helpful comments. To answer these questions, we have firstly provided the best-fitted EXAFS results of the Ru catalysts in Table R5 (also in Table S2 of the supporting information). It could be found that the mean Ru-Ru coordination number of the Ru_n/NC catalyst was 2. By contrast, the Ru-Ru coordination number of a 1 nm pure Ru_n nanocluster in our case (corresponding to a Ru₁₃ nanocluster as shown in Figure R13) was calculated to be 6.5. *Calculation details:* the corner Ru atom numbers of the Ru₁₃ nanocluster were 12 and each corner Ru atom possessed a Ru-Ru coordination number of 6. Thus, the total Ru-Ru coordination numbers of the corner Ru atoms were 12*6=72. By contrast, the center Ru atom of the Ru₁₃ nanocluster possessed a Ru-Ru coordination number of 12, which made the total Ru-Ru coordination numbers of the Ru₁₃ nanocluster as 72+12=84. As such, the average Ru-Ru coordination numbers of the Ru₁₃ nanocluster would be 84/13=6.5.

Since only Ru-Ru and Ru-N coordination were identified in the Ru_n/NC catalyst, the reduction of its Ru-Ru coordination number was most likely derived from the presence of Ru single atoms, because the Ru single atoms were solely coordinated with N and thus decreased the mean Ru-Ru coordination number of the Ru_n/NC catalyst. Careful aberration-corrected HAADF-STEM observation of the Ru_n/NC catalyst in Figure R14 also verified the presence of Ru single atoms in the Ru_n/NC catalyst. On the other hand, the existence of the partially positively charged Ru single atoms in the Ru_n/NC catalyst inevitably increased the average oxidation state of Ru, because of which the edge absorption energy of the Ru_n/NC catalyst showed a significant increase relative to that of the Ru foil as pointed by the Reviewer.

Table R5. The best-fitted EXAFS results of the Ru_n/NC and the Ce₁-Ru_n/NC catalysts.

Sample	Shell	CN	R (Å)	σ^2 (10 ⁻² Å ²)	ΔE_0 (eV)	r-factor (%)
--------	-------	----	-------	---	-------------------	--------------

Ru foil	Ru-Ru	12	2.67	0.4	4.1	1.6
Ru _n /NC	Ru-N	3.8	2.03	0.5	-6.2	1.3
	Ru-Ru	2.3	2.73	2.0	-6.2	
Ce ₁ -Ru _n /NC	Ru-N	4.3	2.03	0.8	-4.3	0.7
	Ru-Ru	2.1	2.67	2.2	-4.3	

CN is the coordination number, R is the average distance for different coordination pairs, σ^2 is the Debye-Waller factor, and ΔE_0 is the inner potential correction. The accuracies of the above parameters are estimated as CN, $\pm 20\%$; R, $\pm 1\%$; σ^2 , $\pm 20\%$; ΔE_0 , $\pm 20\%$. The data range used for data fitting in k-space (Δk) and R-space (ΔR) are 3.0-13.0 \AA^{-1} and 1.0-2.9 \AA for Ru, 3.0-8.0 \AA^{-1} and 1.0-2.0 \AA for Ce, respectively.

Figure R13. (a) The simulated Ru₁₃ nanocluster (corresponding to a Ru_n size of 1 nm). (b) The Ru-Ru coordination numbers of the corner and center Ru atoms in the Ru₁₃ nanocluster.

Figure R14. (a)-(d) The aberration-corrected HAADF-STEM images of the Ru_n/NC catalyst. It was identified that Ru single atoms and small Ru_n nanoclusters coexisted on the NC support in the Ru_n/NC catalyst.

We have also added these contents to the revised manuscript, as read in Line 11, Page 5: “On the other hand, as suggested by the best fitted Fourier transforms of the Ru K-edge extended XAFS (EXAFS) spectrum in Table S2 (corresponding fitting curves were shown in Figure S9-10), the mean Ru-Ru coordination number of the Ru_n/NC catalyst was about 2, which was

markedly lower than that of a pure 1 nm Ru_n nanocluster (6.5, $n=13$) as displayed in Figure S11. This result manifested the copresence of Ru single atoms and Ru_n nanoclusters in the Ru_n/NC catalyst because the Ru single atoms were solely coordinated with N and thus decreased the mean Ru-Ru coordination number of the Ru_n/NC catalyst. Careful aberration-corrected HAADF-STEM observation of the Ru_n/NC catalyst in Figure S12 also revealed the presence of Ru single atoms in the Ru_n/NC catalyst. The partially positively charged Ru single atoms in the Ru_n/NC catalyst inevitably increased the average oxidation state of Ru, because of which the edge absorption energy of the Ru_n/NC catalyst showed a significant increase relative to that of the Ru foil.”

For the second question, we would like to give more detailed description concerning the sentence “It was demonstrated in Figure S8 that electrons could be facily transferred from Ce to Ru with a net electron transfer number of 0.05, which agreed well with the XANES data.” As displayed in Figure 2a, the edge energy of the $\text{Ce}_1\text{-Ru}_n/\text{NC}$ catalyst presented a negative shift compared with that of the Ru_n/NC catalyst, which suggested that Ru component in the $\text{Ce}_1\text{-Ru}_n/\text{NC}$ catalyst was more electron-rich than that in the Ru_n/NC catalyst. Considering the only difference of Ru between the $\text{Ce}_1\text{-Ru}_n/\text{NC}$ catalyst and the Ru_n/NC catalyst was that the $\text{Ce}_1\text{-Ru}_n/\text{NC}$ catalyst contained Ce single atoms, we thus concluded that the mean electron density enhancement of Ru in the $\text{Ce}_1\text{-Ru}_n/\text{NC}$ catalyst was originated from the Ce electron donation in view of the larger electronegativity of Ru (2.2) than Ce (1.1). To check this assumption, we have performed the Bader charge analysis as displayed in Figure S13 in the revised manuscript. It was found that the Ce electrons could be facily transferred to Ru with a net electron transfer number of 0.05, which contributed to the mean electron density enhancement of Ru in the $\text{Ce}_1\text{-Ru}_n/\text{NC}$ catalyst relative to that in the Ru_n/NC catalyst. The result of the mean electron density enhancement of Ru in the $\text{Ce}_1\text{-Ru}_n/\text{NC}$ catalyst as calculated by the Bader charge analysis was in good agreement with corresponding experimental data acquired by the XANES measurement and that was what we want to describe. To avoid the misleading descriptions as pointed by the Reviewer, we have rewritten these contents in the revised manuscript as read in Line 2, Page 6: “It was further found in Figure 2a that the edge energy of the $\text{Ce}_1\text{-Ru}_n/\text{NC}$ catalyst showed a negative shift relative to the Ru_n/NC catalyst and thus an enhancement of its mean Ru electron density,^{26,27} which possibly derived from Ce electron donation as the only difference of Ru between the $\text{Ce}_1\text{-Ru}_n/\text{NC}$ catalyst and the Ru_n/NC catalyst was the presence of Ce single atoms in the $\text{Ce}_1\text{-Ru}_n/\text{NC}$ catalyst. To check this assumption, the Bader charge analysis was further conducted. As suggested in Figure S13, the Ce electrons could be facily transferred to Ru with a net electron transfer number of 0.05, therefore enhancing the mean electron density of Ru in the $\text{Ce}_1\text{-Ru}_n/\text{NC}$ catalyst relative to that of the Ru_n/NC catalyst. The mean electron density enhancement of Ru in the $\text{Ce}_1\text{-Ru}_n/\text{NC}$ catalyst as calculated by the Bader charge analysis was in good agreement with corresponding experimental data acquired by the XANES measurement.”

Comment 3: The authors claim that they have Ru clusters in their main compounds. In the DFT model, these clusters comprise more than 12 Ru atoms. This contradicts the EXAFS fitting, where Ru has, on average, only two neighbours. In a cluster of 12 or more, this number is tremendously larger. Thus, the majority of Ru is not present in the form of the assumed clusters.

Response: Thanks for the Reviewer’s insightful comment. By virtue of careful microscopic observation, uniform Ru_n nanoclusters with an average particle size of about 1 nm were identified in the Ru_n/NC catalysts. To this end, we have built a Ru_{13} nanocluster with a diameter of about 1 nm for simulating the ultrafine Ru_n nanoclusters based on their similar particle size. As pointed by the Reviewer, the Ru-Ru coordination number of the Ru_{13} nanocluster model (calculated to be 6.5, Figure R13) was obviously larger than that obtained by the EXAFS fitted result (with a mean Ru-Ru coordination number of 2). Regarding this problem, we have made detailed discussions in response to Comment 2 and demonstrated that the coexistence of Ru single atoms in the Ru_n/NC catalyst was the reason for its low Ru-Ru coordination numbers. Therefore, to better simulate the Ru_n/NC catalyst, we have reconstructed the structure model

by considering both the particle size of the Ru_n nanocluster (1 nm) and the presence of Ru single atoms as will be discussed in the following part.

We agree with the Reviewer that due to the low Ru-Ru coordination numbers, the relative content of Ru single atoms would be higher than that of the small Ru_n nanoclusters. To obtain the relative ratio of the Ru single atoms to the Ru_n nanoclusters, we have performed the following calculations. Firstly, we defined the number of the Ru single atoms and the Ru_n nanoclusters ($n=13$ in our case) of the Ru_n/NC catalyst as x and y , respectively. Therefore, the total Ru atom numbers of the Ru_n/NC catalyst would be $(x + 13y)$. As mentioned above, the Ru-Ru coordination number of the Ru_{13} nanocluster was 6.5 while the value was 0 for Ru single atoms (with only Ru-N coordination), which meant that the total Ru-Ru coordination numbers of the Ru_n/NC catalyst were $(x*0 + 13y*6.5)$. Therefore, we could acquire the average Ru-Ru coordination number of the Ru_n/NC catalyst through dividing its total Ru-Ru coordination numbers by its total Ru atom numbers as illustrated in equation 1 below.

$$N_A = \frac{\text{Total Ru-Ru coordination numbers}}{\text{Total Ru atom numbers}} = \frac{(x * 0 + 13y * 6.5)}{(x + 13y)}$$

N_A represents for the average Ru-Ru coordination number in the Ru_n/NC catalyst;

x denotes the number of the Ru single atoms in the Ru_n/NC catalyst;

y indicates the number of the Ru_n nanoclusters in the Ru_n/NC catalyst;

As unveiled by the aforementioned EXAFS fitting results, the average Ru-Ru coordination number (N_A) of the Ru_n/NC catalyst was 2. For simplification, we have chosen one Ru_n nanocluster in the Ru_n/NC catalyst as the studying object, corresponding to a y value of 1. As such, the relative ratio of the Ru single atoms to the Ru_n nanoclusters in the Ru_n/NC catalyst was calculated to be 29:1 according to the equation 1 (based on $N_A=2$ and $y=1$).

Figure R15. The Gibbs free energy diagrams for water dissociation over the Ru single atom catalyst (Ru_1/NC).

However, in spite of the large number of Ru single atoms in the Ru_n/NC catalyst, these isolated Ru atoms were primarily insufficient for dissociating water molecule (the rate-determining step of alkaline HER). As demonstrated in Figure R15, the Gibbs free energy barrier for water dissociation over the Ru single atom was as huge as 1.03 eV. In addition, water dissociation over the Ru single atom was a thermodynamically unfavorable endothermic process with an endothermic energy of 0.67 eV. These results suggested the low reactivity of the Ru single atoms for the alkaline HER. To further confirm this, we have synthesized a pure Ru single atom catalyst (Ru_1/NC , the synthesis details were provided in the methods part of the main text) and the formation of Ru single atoms on the NC support was verified by careful

aberration-corrected HAADF-STEM observation (Figure R16). Corresponding alkaline HER evaluation results (LSV curve) in Figure R17 showed that the catalytic activity of the Ru₁/NC catalyst was much lower than that of the Ru_n/NC catalyst, which further revealed the low alkaline HER activity of the Ru single atoms.

Figure R16. (a)-(f) The aberration-corrected HAADF-STEM images of the Ru₁/NC catalyst with varied magnifications.

Figure R17. The LSV curves for the Ru₁/NC catalyst and the Ru_n/NC catalyst during the alkaline HER evaluations. The Ru loading amounts of the Ru₁/NC catalyst and the Ru_n/NC catalyst were 0.2wt.% and 1.2wt.%, respectively.

Actually, a common practice by previous reports (*Adv. Funct. Mater.*, 2023, 33, 2213058; *Appl. Catal., B*, 2022, 312, 121378; *Small*, 2021, 17, 2101163) for simulating the catalyst with coexisted Ru single atoms and Ru nanoclusters/nanoparticles was to build a dual Ru₁-Ru_n structure model. Therefore, we have further constructed a dual Ru₁-Ru₁₃ structure model as shown in Figure R18 to simulate our Ru_n/NC catalyst by considering both the particle size of the Ru_n nanocluster (about 1 nm, corresponding to n=13) and its neighboring Ru single atoms. The Ru-N coordination numbers for the Ru single atom and for the Ru₁₃ nanocluster were both

four according to the EXAFS fitting results. As indicated in Figure R19, the dual Ru₁-Ru₁₃ site showed a quite low Gibbs free energy barrier (0.49 eV) for water dissociation and meanwhile the process was exothermic by 1.42 eV, showing a favorable reaction thermodynamics. These results revealed the good alkaline hydrogen evolution activity of the dual Ru₁-Ru₁₃ site and made it a reasonable structural model for simulating the Ru_n/NC catalyst.

We have also added these contents to the revised manuscript as read in Line 21, Page 11: “In view of the copresence of Ru single atoms and Ru_n nanoclusters in the Ru_n/NC catalyst, we have built a dual Ru₁-Ru₁₃ model for simulating the Ru_n/NC catalyst by considering both the particle size of the Ru_n nanocluster (1 nm, Figure S11) and its neighboring Ru single atoms as was also commonly employed by previous reports for simulating the united catalyst with coexisted Ru single atoms and Ru nanoclusters/nanoparticles.^{23,36,37} All the structure details of the constructed models were provided in supporting information (Figure S31-34). To begin with, it should be noted that pure Ru single atoms on the NC support were insufficient for dissociating water molecule that was regarded as the rate-determining step of alkaline HER. As demonstrated in Figure S35, the Gibbs free energy barrier for water dissociation over the Ru single atom was as huge as 1.03 eV, indicating its low alkaline HER activity. To further examine the alkaline HER activity of Ru single atoms, we have synthesized a Ru₁/NC control (the synthetic details were demonstrated in methods part). The formation of Ru single atoms in the Ru₁/NC catalyst was confirmed by the aberration-corrected HAADF-STEM images of it in Figure S36. As further revealed by the alkaline HER evaluation results (LSV curve) in Figure S37, the catalytic activity of the Ru₁/NC catalyst was much lower than that of the Ru_n/NC catalyst. By contrast, the dual Ru₁-Ru₁₃ sites presented a quite low Gibbs free energy barrier for dissociating water molecules compared with that of the pure Ru single atoms (Figure S1 and Figure S35), making the dual Ru₁-Ru₁₃ site a reasonable structural model for simulating the Ru_n/NC catalyst.”

Figure R18. (a) Side view and (b) top view of the dual Ru₁-Ru₁₃ structural model.

Figure R19. The Gibbs free energy diagrams for water dissociation over the dual Ru₁-Ru₁₃ sites. The insets of the picture are corresponding schematic illustrations for each step.

Comment 4: The theoretical calculations do not align with the experimental observations. 1. The clusters are too large compared to the EXAFS coordination number. Figure 5e implies an overpotential of almost 400 mV, which does not fit to the experimentally observed one at all. Therefore, the DFT analysis is without any meaning for the real system.

Response: We thank the Reviewer for these important comments. As pointed by the Reviewer, the clusters are too large compared to the EXAFS coordination number. For this question, we have provided detailed explanation in response to Comment 2 and Comment 3 and unveiled that the copresence of Ru single atoms was the reason for the low Ru-Ru coordination numbers of the Ru_n/NC catalyst. To better simulate the Ru_n/NC catalyst, we have reconstructed the model for the Ru_n/NC catalyst by taking into account of both the particle size of the Ru_n nanoclusters (about 1 nm) and the coexistence of Ru single atoms. The rationale of the reconstructed Ru₁-Ru₁₃ model for simulating the Ru_n/NC catalyst was also detailedly discussed in the response to Comment 3.

Figure R20. The Gibbs free energy diagrams for water dissociation over the Ce₁/NC catalyst. The insets of the picture are corresponding schematic illustrations for each step.

As far as the Ce₁-Ru_n/NC catalyst was concerned, since the Ru single atoms and the Ce single atoms therein were both insufficient for dissociating water in terms of their huge Gibbs free energy barriers for water dissociation as demonstrated in Figure R15 (1.03 eV for Ru₁) and

Figure R20 (0.8 eV for Ce₁), we have further examined the water dissociation properties of the dual Ru₁-Ru₁₃ sites and the dual Ce₁-Ru₁₃ sites in the Ce₁-Ru_n/NC catalyst by the DFT calculations. Keeping in mind that the experimental test results in Figure 4 demonstrated that the alkaline HER activity of the Ce₁-Ru_n/NC catalyst was much higher than the Ru_n/NC catalyst, the dual Ce₁-Ru₁₃ sites might be much more reactive than the dual Ru₁-Ru₁₃ sites because it was the sole difference between the Ce₁-Ru_n/NC catalyst and the Ru_n/NC catalyst.

As indicated in Figure R21, water dissociation over the dual Ce₁-Ru₁₃ sites was highly exothermic by 4.5 eV, which was 3.2 times that of the dual Ru₁-Ru₁₃ sites (1.42 eV). This result unveiled that the dual Ce₁-Ru₁₃ sites were much more thermodynamically favorable to dissociate water molecules (rate-determining step of alkaline HER) than the dual Ru₁-Ru₁₃ sites. On the other hand, the water dissociation energy barrier over the dual Ce₁-Ru₁₃ sites (0.1 eV) was significantly lower than that over the dual Ru₁-Ru₁₃ sites (0.49 eV). These results disclosed the excellent catalytic reactivity of the dual Ce₁-Ru₁₃ sites for alkaline HER compared with the dual Ru₁-Ru₁₃ sites, and also suggested that it was reasonable to use the dual Ce₁-Ru₁₃ structural model (Figure R22) to simulate the highly efficient Ce₁-Ru_n/NC catalyst. All of these contents are now added to the revised manuscript as read in Line 19, Page 12: “As far as the Ce₁-Ru_n/NC catalyst was concerned, both the Ru single atoms and Ce single atoms in it were insufficient for dissociating water in terms of their huge Gibbs free energy barriers for water dissociation as demonstrated in Figure S35 and Figure S38, respectively. Keeping in mind that the experimental test results in Figure 4 demonstrated that the alkaline HER activity of the Ce₁-Ru_n/NC catalyst was much higher than the Ru_n/NC catalyst, it was thus reasonable to use the dual Ce₁-Ru₁₃ model for simulating the Ce₁-Ru_n/NC catalyst because it was the only difference between the Ce₁-Ru_n/NC catalyst and the Ru_n/NC catalyst.” Line 1, Page 13, “The DFT calculations suggested that the exothermic energy of the dual Ce₁-Ru₁₃ sites (4.5 eV) was 3.2 times that of the dual Ru₁-Ru₁₃ sites (1.42 eV) in water activation process, which unveiled that the dual Ce₁-Ru₁₃ sites were more thermodynamically favorable to dissociate water molecules than the dual Ru₁-Ru₁₃ sites. In addition, the water dissociation energy barrier of the dual Ce₁-Ru₁₃ sites (0.1 eV) was also markedly lower than that of the dual Ru₁-Ru₁₃ sites (0.49 eV). Therefore, the dual Ce₁-Ru₁₃ sites catalyst were both more thermodynamically and kinetically beneficial to promote water dissociation relative to the dual Ru₁-Ru₁₃ sites.”

Figure R21. The Gibbs free energy diagrams for water dissociation over the dual Ce₁-Ru₁₃ site and over the dual Ru₁-Ru₁₃ site. Corresponding schematic illustrations for each step over the

dual Ce₁-Ru₁₃ site and over the dual Ru₁-Ru₁₃ site were also provided below the Gibbs free energy diagram.

Figure R22. (a) Side view and (b) top view of the dual Ce₁-Ru₁₃ structural model.

The Reviewer also mentioned the difference between the overpotentials obtained through the theoretical calculations and the experiment measurements. Concerning this problem, we would like to make a detailed explanation. In current work, the H adsorption energy as shown in Figure 5e was calculated by the “computational hydrogen electrode model” that was proposed by J. K. Nørskov, et al. in 2005 (*J. Electrochem. Soc.* 2005, 152 (3), J23). By virtue of its robustness and relatively easy operation, the “computational hydrogen electrode model” was the mostly used method for calculating H adsorption energy of electrocatalyst at present. However, this method was greatly affected by the polarization effect during the calculations. As a result, though the calculated reaction energy reflected the reasonable reaction trend, the value of the calculated reaction energy commonly displayed an evident deviation from the experiment test result. To overcome this disadvantage, in the revised manuscript, we have further employed a new “CANDEL implicit solvation model” method that could mediate the polarization effect to calculate the H adsorption energies. During the calculations, the structures during reaction were fully relaxed until the final force on each atom was less than 0.01 eV Å⁻¹. The H adsorption energies on the structure models of the dual Ce₁-Ru₁₃ site and the dual Ru₁-Ru₁₃ site were obtained by adding the vibrational contribution of H to the electronic energy of corresponding reaction systems. As displayed in Figure R23, the calculated H adsorption energies of the dual Ce₁-Ru₁₃ site and the dual Ru₁-Ru₁₃ site via the “CANDEL implicit solvation model” method were -0.096 eV and -0.142 eV, respectively, which were both closer to the experimentally measured onset potentials of them than that calculated through the “computational hydrogen electrode model” method. We have also added these contents to the revised manuscript as read in Line 24, Page 13: “In another respect, the calculated H adsorption energies of the dual Ce₁-Ru₁₃ site and the dual Ru₁-Ru₁₃ site via the “computational hydrogen electrode model” method³⁹ were obviously larger than that of the experimentally measured onset potentials of them. This result was due to the fact that the reaction system was a grand canonical ensemble in the computational hydrogen electrode model, which was significantly affected by the polarization effect. As a consequence, despite that the calculated reaction energy suggested the reasonable reaction trend, the value of the calculated reaction energy commonly displayed a marked deviation from the experiment test result. Therefore, we have further performed the H adsorption energy calculations for the dual Ce₁-Ru₁₃ site and the dual Ru₁-Ru₁₃ site taking the “CANDEL implicit solvation model” method. As indicated in Figure S39, the calculated H adsorption energies on the dual Ce₁-Ru₁₃ site and on the dual Ru₁-Ru₁₃ site using this method were -0.096 eV and -0.142 eV, respectively, which were both closer to the experimentally measured onset potentials of them than that calculated through the “computational hydrogen electrode model” method.”

Figure R23. The H adsorption energies of the dual Ce₁-Ru₁₃ site and the dual Ru₁-Ru₁₃ site calculated by the CANDEL implicit solvation model method.

Comment 5: The plot of the long-term stability measurement (Figure 4f) is missing ticks on the right y-axis to make it easy to read how much the sample deactivates. Using a ruler, I could find out that there is a substantial deactivation in the range of 50-100 mV. If the authors want to know if their material would be stable under industrial conditions, they must measure it at 80 °C and a current density of 400 mA/cm².

Response: Motivated by the Reviewer's kind suggestion, we have conducted the stability measurements at 80 °C and a current density of 400 mA/cm² on an alkaline anion-exchange-membrane water electrolysis (AEMWE) device using the Ce₁-Ru_n/NC as the cathodic catalyst and the nickel foam as the anodic catalyst. The obtained chronopotentiometry curve was presented in Figure R24 and we have added ticks on the right y-axis to make it easy to read how much the sample deactivates following the Reviewer's kind suggestion. It was found that the Ce₁-Ru_n/NC catalyst was quite stable during the stability test under the AEMWE conditions for 100 hours without obvious deactivation.

Figure R24. The chronopotentiometry curve of the Ce₁-Ru_n/NC catalyst under AEMWE conditions at a reaction temperature of 80 °C and a current density of 400 mA/cm². The inset of Figure R24 was the photograph of the AEMWE device.

We have also included these contents in the revised manuscript as read in Line 18, Page 10: “To examine the practical application potential of the Ce₁-Ru_n/NC catalyst, we have further measured the durability of the Ce₁-Ru_n/NC catalyst on an alkaline anion-exchange-membrane water electrolysis (AEMWE) device using the Ce₁-Ru_n/NC as the cathodic catalyst and the

nickel foam as the anodic catalyst. The chronopotentiometry curve acquired by setting the reaction temperature at 80 °C and the current density of 400 mA/cm² in Figure 4f suggested that the Ce₁-Ru_n/NC catalyst displayed quite good stability for 100 hours testing.”

Comment 6: The general statement that the industry would prefer alkaline electrolysis is not true and is an oversimplification. It should be changed. Alkaline and PEM both have their advantages for industrial applications.

Response: We thank the Reviewer for this very helpful comment and apologize for the misleading descriptions. To more clearly describe the contents as pointed by the Reviewer, we have rewritten the sentence in the revised manuscript as read in Line 26, Page 1: “**In practical applications, alkaline water electrolysis and proton exchange membrane (PEM)-based water electrolysis in acid electrolyte both have their advantages for producing hydrogen as was well reviewed by previous reports.³ In a typical PEM system, a proton exchange membrane was used as solid electrolyte by means of which proton could be facily transferred to cathode,⁴ enabling fast hydrogen evolution kinetics. However, unlike the direct proton supply in acid HER, the proton was provided by the dissociation of H₂O during the alkaline HER (Volmer step, equation 1).⁵⁻⁷ The substantial energy barrier for the cleavage of OH-H bond and sluggish supply of proton significantly impeded the reaction rate of alkaline HER.⁸⁻¹⁰ Even for the commercial Pt/C electrocatalysts, about two orders of magnitude reduction in HER activity was commonly identified when used in alkaline electrolyte.¹¹ To this end, promoting electrocatalytic water dissociation over catalyst in alkaline electrolyte became of paramount importance to boost corresponding HER activity.”**

Comment 7: Replacing Platinum by Ruthenium is not a real advantage as they are similarly rare in the earth's crust.

Response: We do agree with the Reviewer that both platinum and ruthenium are of low contents in the earth's crust. In spite of that, the ruthenium metal has two distinct advantages over the platinum metal toward the alkaline HER: (i) the metal price of ruthenium (ca. 15 \$ g⁻¹, Sept 2023) was less than half that of platinum (ca. 34 \$ g⁻¹, Sept 2023), making the ruthenium a more attractive candidate for alkaline HER in view of cost. (ii) the ruthenium exhibited a much lower energy barrier for water dissociation relative to that of the platinum as displayed in Figure R25 (adapted from J. Am. Chem. Soc. 2016, 138, 49, 16174-16181). This in turn largely promoted the water dissociation (rate-determining step of alkaline HER) and thus improved the alkaline HER efficiency, which also potentially reduced the usage amount of Ru metal. We have added these contents to the revised manuscript as read in Line 4, Page 2: “**In the past decades, Ru has shown great potential to substitute the Pt for alkaline HER by virtue of two-fold: (i) the metal price of Ru (ca. 15 \$ g⁻¹, Sept 2023) was less than half that of the Pt (ca. 34 \$ g⁻¹, Sept 2023); and (ii) the energy barrier for water dissociation over Ru was much lower than that over Pt, which largely promoted the water dissociation and thus improved the overall alkaline HER efficiency, also potentially reducing the usage amount of Ru.¹²⁻¹⁴”**

Figure R25. The Gibbs free energy diagram of HER on Ru and Pt, respectively (adapted from the J. Am. Chem. Soc. 2016, 138, 49, 16174-16181).

Comment 8: Tafel slopes cannot be reliably determined from potentiodynamic methods. They must be measured under steady-state conditions with various CA measurements (see 10.1021/acseenergylett.1c00608 or /10.1016/j.mtener.2022.101123).

Response: We thank the Reviewer for this very important comment. As suggested by the Reviewer, we have remeasured the Tafel slopes of the catalysts in the revised manuscript using the potentiodynamic methods. It was demonstrated in Figure R26 (corresponding CA curves were also shown in Figure R27) that the remeasured Tafel slope values of the Ce₁-Ru_n/NC catalyst, the 20wt.% Pt/C catalyst, the Ru_n/NC catalyst, and the Ru_n-CeO₂/NC catalyst were 41.5 mV dec⁻¹, 60.6 mV dec⁻¹, 134.4 mV dec⁻¹, and 180.7 mV dec⁻¹, respectively. It was evident that the Tafel slope value of the Ce₁-Ru_n/NC catalyst was the lowest among these catalysts, which suggested the fast reaction kinetics for hydrogen evolution over the Ce₁-Ru_n/NC catalyst. We have also updated these results in the revised manuscript as read in Line 26, Page 8: “We have also measured the Tafel slopes of these catalysts by virtue of the potentiodynamic method as suggested previously.^{33,34} It was demonstrated in Figure 4c that the obtained Tafel slope value of the Ce₁-Ru_n/NC catalyst (41.5 mV dec⁻¹) was markedly lower than that of the 20wt.% Pt/C catalyst (60.6 mV dec⁻¹), the Ru_n/NC catalyst (134.4 mV dec⁻¹), and the Ru_n-CeO₂/NC catalyst (180.7 mV dec⁻¹), indicating fast reaction kinetics for hydrogen evolution over the Ce₁-Ru_n/NC catalyst.”

Figure R26. The Tafel slope values of the Ce₁-Ru_n/NC catalyst, the 20wt.% Pt/C catalyst, the Ru_n/NC catalyst, and the Ru_n-CeO₂/NC catalyst, measured by the potentiodynamic method.

Figure R27. Corresponding CA curves for the Ce_1-Ru_n/NC catalyst, the 20wt.% Pt/C catalyst, the Ru_n/NC catalyst, and the Ru_n-CeO_2/NC catalyst, during the calculation of the Tafel slopes of them by the potentiodynamic method.

REVIEWERS' COMMENTS

Reviewer #2 (Remarks to the Author):

The manuscript has revised based on the comments, providing the additional experimental and theoretical results. The data and analysis adequately test the hypothesis and support their novelty. But, the language and description aspects of scholarly presentation is still poor. The manuscript structure including the abstract and conclusion require further improvement to be published on high-level Nature Communications.

For example, the HER should be written by the hydrogen evolution reaction, at first in the abstract part. In addition, they illustrate the improved Ru atom efficiency by the strong oxophilicity including the abstract part. However, as shown in the results of ICP-OES, the Ce1-Ru/NC shows the similar Ru loading to that of the Ru1-Ru/NC, which further require adequate description. Figure S28 shows the LSV curves and the TOF of catalysts, but the author illustrates the LSV curves and the mass activity curves at page 10 line 314. In addition, they assume the strengthening of the Raman vibration peaks (1533 and 1390 cm^{-2}) is related to OH effects during the alkaline HER. This assumption needs to be supported with the related references. Check and improve the language, structure, figure notation, typo and spacing through the manuscript and SI.

Reviewer #3 (Remarks to the Author):

The authors have satisfactorily addressed most of my previous comments and thus this manuscript is recommended for publication. The authors should note that potentiodynamic means that the Tafel slopes are measured from LSV or CV and steady-state means that they are measured using CA or CP measurements, where the current has stabilized. Please check the usage of these terms in the manuscript again.

Response to the Reviewers' comments:

Reviewer 2

Overall comments: The manuscript has revised based on the comments, providing the additional experimental and theoretical results. The data and analysis adequately test the hypothesis and support their novelty. But, the language and description aspects of scholarly presentation is still poor. The manuscript structure including the abstract and conclusion require further improvement to be published on high-level Nature Communications.

Response: We are very grateful for the Reviewer's highly positive assessment of our revision. As suggested by the Reviewer, in the revised manuscript, we have carefully modified the manuscript structure and rewritten the abstract and conclusion. Please see below our point-by-point responses to the Reviewer's concerns.

Comment 1: For example, the HER should be written by the hydrogen evolution reaction, at first in the abstract part.

Response: Thanks a lot for the Reviewer's kind suggestion. In the updated abstract part, we have revised the word "HER" to "hydrogen evolution reaction (HER)" to make it clear.

Comment 2: In addition, they illustrate the improved Ru atom efficiency by the strong oxophilicity including the abstract part. However, as shown in the results of ICP-OES, the Ce₁-Ru_n/NC shows the similar Ru loading to that of the Ru₁-Ru_n/NC, which further require adequate description.

Response: Thanks for the Reviewer's valuable comment and kind advice. As suggested by the Reviewer, we have included more clear descriptions on the improvement of Ru atom efficiency of the Ce₁-Ru_n/NC catalyst relative to that of the Ru₁-Ru_n/NC catalyst. As read in in Line 10, Page 10, "Moreover, in spite of the similar Ru loadings between the Ce₁-Ru_n/NC catalyst and the Ru₁-Ru_n/NC catalyst, the mass activity normalized to per milligram of Ru of the Ce₁-Ru_n/NC catalyst was markedly higher than that of the Ru₁-Ru_n/NC catalyst as displayed in Figure S29, which unveiled the higher Ru atom efficiency for alkaline hydrogen evolution over the Ce₁-Ru_n/NC catalyst than that over the Ru₁-Ru_n/NC catalyst."

Comment 3: Figure S28 shows the LSV curves and the TOF of catalysts, but the author illustrates the LSV curves and the mass activity curves at page 10 line 314.

Response: Thanks for the Reviewer's kind reminding and we sincerely apologize for the typo. In the updated manuscript and supporting information, we have revised the "mass activity curves" to "TOF values" with reference to Figure S28 accordingly.

Comment 4: In addition, they assume the strengthening of the Raman vibration peaks (1533 and 1390 cm⁻²) is related to OH effects during the alkaline HER. This assumption needs to be supported with the related references.

Response: Thanks for the Reviewer's helpful suggestions. However, as we mentioned in the main text, after extensive paper searching, we found that there was no record of the two Raman vibration peaks of the Ce₁-Ru_n/NC catalyst at 1533 cm⁻¹ and 1390 cm⁻¹. Therefore, we used the Gaussian fitting method to examine the two peaks' assignment as commonly employed previously. It was found that the Raman vibration peaks of the Ce₁-Ru_n/NC catalyst at 1533 cm⁻¹ and 1390 cm⁻¹ were assigned to the Ce-N stretching vibrations in the Ce₁-Ru_n/NC catalyst. It was also identified that when introducing OH to the Ce-N sites of the Ce₁-Ru_n/NC catalyst, corresponding Raman peak intensities at the 1533 cm⁻¹ and 1390 cm⁻¹ were significantly strengthened, which was in good agreement with the *in situ* Raman spectra measurement results as displayed in Figure 4c. As such, both the Gaussian fitting results and the experiment data of the *in situ* Raman measurement revealed the OH strengthening effect on the Raman vibration peaks of the Ce₁-Ru_n/NC catalyst at the 1533 cm⁻¹ and 1390 cm⁻¹.

Comment 5: Check and improve the language, structure, figure notation, typo and spacing through the manuscript and SI.

Response: Thanks for the Reviewer's valuable comment and kind advice. As suggested by the Reviewer, we have carefully improved the language, structure, figure notation, typo and spacing through the manuscript and SI. The main changes are listed below:

1. The abstract part was rewritten, wherein the word "HER" was revised to "hydrogen evolution reaction (HER)"
2. In Line 26, Page 1, the word "approach" was revised to "approaches"
3. In Line 30, Page 1, the sentence "as was well reviewed by previous reports" was deleted.
4. In Line 33, Page 1, these words "for scissoring the OH-H bond" were added.
5. In Line 38, Page 1, the word "electrocatalytic" was deleted.
6. In Line 8, Page 2, the sentence "which largely promoted the water dissociation and thus improved the overall alkaline HER efficiency, also potentially reducing the usage amount of Ru" was deleted.
7. In Line 7, Page 2, the word "firstly" was added to make it clear of the development of Ru-based alkaline HER catalyst.
8. In Line 9, Page 2, these words "its substantial uphill energy" were revised to "their substantial uphill energies".
9. In Line 10, Page 2, these words "its alkaline HER activity" were revised to "their alkaline HER activities".
10. In Line 17, Page 2, the word "the" was added.
11. In Line 8, Page 3, these words "over the Ce₁-Ru_n/NC catalyst" were added.
12. In Line 10, Page 3, the sentence "As such, alkaline HER activity of the Ce₁-Ru_n/NC catalyst was largely improved, providing a new avenue to more effective alkaline hydrogen evolution." was added.
13. In Line 17, Page 3, these words "the treatment of" were added.
14. In Line 21, Page 3, the sentence "to the synthesis of the Ce₁-Ru_n/NC catalyst except for the absence of Ce" was deleted.
15. In Line 22, Page 3, the word "further" was revised to "also"
16. In Line 34, Page 3, the sentence "also confirmed the uniform dispersion of Ce species in it." was added.
17. In Line 1, Page 4, the sentence "suggesting the copresence of Ru nanoclusters and Ce single atoms in it." was added.
18. In Line 2, Page 4, these words "for both of" were added.
19. In Line 3, Page 4, the sentence was modified to "because of the ultrasmall Ru nanoclusters in them"
20. In the caption of Figure 2, the sentence "It should be noted that the colors of these spheres in (a,h,o) are in line with that of Figure 1." was added to make it clear.
21. In Line 21, Page 4, the sentence was revised to "which demonstrated their oxidation states of Ru were between 0 and +4."
22. In the caption of Figure 3, the sentence "The unit of a.u. indicated the arbitrary units (hereafter)." was added to make it clear.
23. In Line 22, Page 5, the word "metallic" was added.
24. In Line 1, Page 6, the sentence was modified to "thus an enhancement of the mean Ru

electron density of the Ce₁-Ru_n/NC catalyst”

25. In Line 10, Page 6, the sentence “The mean electron density enhancement of Ru in the Ce₁-Ru_n/NC catalyst as calculated by the Bader charge analysis was in good agreement with corresponding experimental data acquired by the XANES measurement.” was deleted.

26. In Line 20, Page 6, the sentence was modified to “The enhanced Ru-N coordination ratios in the Ce₁-Ru_n/NC and Ru_n/NC were also supported by the EXAFS wavelet transform results.”

27. In Line 24, Page 6, the sentence was revised to “On the other hand, the Ce L_{III}-edge XANES and fitted EXAFS spectra of the Ce₁-Ru_n/NC and the reference CeO₂ in Figure 3f-g and Figure S14-15 further confirmed the single atom nature of Ce species in the Ce₁-Ru_n/NC catalyst”

28. In Line 11, Page 7, the sentence “before the electrochemical alkaline HER evaluations” was deleted.

29. In Line 14, Page 7, the sentence was revised to “In addition, the OH desorption peak of the Ce₁-Ru_n/NC catalyst also displayed a negative shift relative to that of the Ru_n/NC catalyst that again verified the stronger OH adsorption on the Ce₁-Ru_n/NC catalyst than that on the Ru_n/NC catalyst.”

30. In Line 25, Page 7, the sentence “the instrument model was schematically illustrated in Figure S23” was included.

31. In the caption of Figure 4, the sentence “To be noted, the colors of these spheres in (b,e,f) are in line with that of Figure 1.” was added to make it clear.

32. In Line 24, Page 8, the sentence was revised to “We have also measured the Tafel slopes of these catalysts under steady-state conditions using the CA measurements as suggested previously.”

33. In Line 4, Page 9, these words “among these catalysts” were added.

34. In Line 5, Page 9, the sentence was revised to “we have further conducted the turnover frequency (TOF) measurements for these catalysts in the potential range from -0.01 V to -0.06 V.”

35. In the caption of Figure 5, the sentence “The colors of these spheres in (e) are in line with that of Figure 1.” was added to make it clear.

36. In Line 9, Page 10, we have revised the “mass activity curves” to “TOF values” accordingly.

37. In Line 10, Page 10, the sentence “Moreover, in spite of the similar Ru loadings between the Ce₁-Ru_n/NC catalyst and the Ru₁-Ru_n/NC catalyst, the mass activity normalized to per milligram of Ru of the Ce₁-Ru_n/NC catalyst was markedly higher than that of the Ru₁-Ru_n/NC catalyst as displayed in Figure S29, which unveiled the higher Ru atom efficiency for alkaline hydrogen evolution over the Ce₁-Ru_n/NC catalyst than that over the Ru₁-Ru_n/NC catalyst.” was added to show the higher Ru atom efficiency of the Ce₁-Ru_n/NC catalyst than that of the Ru₁-Ru_n/NC catalyst.

38. In the caption of Figure 6, the sentence “The red sphere, white sphere, yellow sphere, light brown sphere, brown sphere and silvery sphere denote O, H, Ce, Ru, C, and N, respectively.” was added to clarify the pictures in Figure 6f.

39. In Line 40, Page 13, the sentence was revised to “which would largely weaken the hydrogen binding strength on the dual Ce₁-Ru₁₃ site for more favorable hydrogen production.”

40. In Line 43, Page 13, the conclusion part was rewritten and all of the updated contents were marked in red.

41. The “Details of theoretical calculations” was added to the methods part to make clear the details of the theoretical calculations.

42. In addition, we have also updated the supporting information. To make it clear, all the

caption of figures were carefully checked and the typo and spacing were corrected accordingly. Moreover, we have also added the references (Ref 2 to Ref 116) previously incorporated in Table S4 and S5 to the supporting reference part to make it more readable.

Reviewer 3

Overall comments: The authors have satisfactorily addressed most of my previous comments and thus this manuscript is recommended for publication.

Response: We are very grateful for the Reviewer's highly positive evaluation of our revision.

Comment 1: The authors should note that potentiodynamic means that the Tafel slopes are measured from LSV or CV and steady-state means that they are measured using CA or CP measurements, where the current has stabilized. Please check the usage of these terms in the manuscript again.

Response: Thanks for the Reviewer's kind reminding and in the revised manuscript, we have carefully checked the use of these terms as mentioned by the Reviewer. To make it clear, we have rewritten the sentence in Line 24, Page 8 to “**We have also measured the Tafel slopes of these catalysts under steady-state conditions using the CA measurements as suggested previously.**^{33,34}”